# Duodenal bacterial proteolytic activity determines sensitivity to dietary antigen through protease-activated receptor-2

Alberto Caminero[1], Justin L. McCarville[1], Heather J. Galipeau[1], Celine Deraison[2], Steve P. Bernier[1], Marco Constante[1], Corinne Rolland[2], Marlies Meisel[3,12], Joseph A. Murray[4], Xuechen B. Yu [5,6], Armin Alaedini[5,6], Brian K. Coombes[7], Premysl Bercik[1], Carolyn M. Southward[1], Wolfram Ruf [8,9], Bana Jabri[3], Fernando G. Chirdo[10], Javier Casqueiro[11], Michael G. Surette[1,7], Nathalie Vergnolle [2] & Elena F. Verdu[1]

Microbe-host interactions are generally homeostatic, but when dysfunctional, they can incite food sensitivities and chronic diseases. Celiac disease (CeD) is a food sensitivity characterized by a breakdown of oral tolerance to gluten proteins in genetically predisposed individuals, although the underlying mechanisms are incompletely understood. Here we show that duodenal biopsies from patients with active CeD have increased proteolytic activity against gluten substrates that correlates with increased Proteobacteria abundance, including *Pseudomonas*. Using *Pseudomonas aeruginosa* producing elastase as a model, we show gluten-independent, PAR-2 mediated upregulation of inflammatory pathways in C57BL/6 mice without villus blunting. In mice expressing CeD risk genes, *P. aeruginosa* elastase synergizes with gluten to induce more severe inflammation that is associated with moderate villus blunting. These results demonstrate that proteases expressed by opportunistic pathogens impact host immune responses that are relevant to the development of food sensitivities, independently of the trigger antigen.

[1] Farncombe Family Digestive Health Research Institute, McMaster University, Hamilton L8S 4K1 ON, Canada. [2] IRSD, Université de Toulouse, INSERM, INRA, ENVT, UPS, Toulouse 31300, France. [3] Department of Medicine, University of Chicago, Chicago 60637 IL, USA. [4] Division of Gastroenterology and Hepatology, Department of Immunology, Mayo Clinic College of Medicine, Rochester 55905 MN, USA. [5] Department of Medicine, Columbia University, New York 10032 NY, USA. [6] Celiac Disease Center, Columbia University, New York 10032 NY, USA. [7] Department of Biochemistry and Biomedical Sciences, McMaster University, Hamilton L8S 4K1 ON, Canada. [8] Center for Thrombosis and Hemostasis, Johannes Gutenberg University Medical Center, Mainz 55131, Germany. [9] Department of Immunology and Microbiology, The Scripps Research Institute, La Jolla 92037 CA, USA. [10] Instituto de Estudios Inmunologicos y Fisiopatologicos - IIFP (UNLP-CONICET). Facultad de Ciencias Exactas, Universidad Nacional de La Plata, La Plata 1900, Argentina. [11] Department of Microbiology, Universidad de Leon, Leon 24071, Spain. [12] Present address: Department of Immunology, University of Pittsburgh School of Medicine, Pittsburgh PA 15213 PA, USA. These authors contributed equally: Alberto Caminero, Justin L. McCarville, Heather J. Galipeau. Correspondence and requests for materials should be addressed to E.F.V. (email: verdue@mcmaster.ca)

The human gastrointestinal tract contains proteases that exert different and complex functions, including cell signaling, immune and barrier functions, and metabolism of dietary components[1,2]. The release, activity, and degradation of host proteases are tightly regulated, and imbalances in these processes is associated with functional and chronic inflammatory conditions, such as irritable bowel syndrome and inflammatory bowel disease[3–6]. Gut bacteria and enteric opportunistic pathogens produce a large array of proteases that can degrade dietary components as well as host proteins, with the potential to impact host immune and functional pathways, leading to pro-inflammatory responses when dysfunctional[7–10]. However, our knowledge of the specific underlying microbe–dietary interactions through which proteases are conducive to inflammation is limited.

One of the most abundant sources of protein in the Western diet is gluten, the general name given to a complex mixture of highly immunogenic and difficult to digest glutamine and proline-rich storage proteins in wheat and related cereals[11]. In humans carrying specific risk alleles, gluten can trigger the common food sensitivity celiac disease (CeD). Recent epidemiological studies have demonstrated the growing seroprevalence, to 1.4% worldwide, of this inflammatory condition affecting the small intestine[12,13]. CeD is characterized by the development of gluten-specific T-cells, anti- tranglutaminase 2 antibodies, and an increase of intraepithelial lymphocytes (IELs)[14]. Because risk alleles (HLA-DQ2 or -DQ8) are necessary, but insufficient to develop disease, additional environmental factors of microbial origin have been proposed[15–17]. In line with this, we have recently shown that undigested gluten can be metabolized by certain opportunistic pathogens, such as Pseudomonas aeruginosa, leading to the production of peptides that more easily translocate across the mucosal barrier with retained immunogenicity[18]. However, it is unknown whether microbial proteolytic activity in the small intestine directly influences host pathways important for the maintenance of homeostasis towards dietary proteins.

Here, we demonstrate that duodenal biopsies from patients with CeD have increased proteolytic activity against gluten peptides which correlates with Proteobacteria abundance. Using a model Proteobacterium, P. aeruginosa, which expresses the gluten-degrading protease elastase (LasB), and its isogenic non-functional lasB mutant[19], we show an elastase-dependent inflammatory response mediated by the protease-activated receptor-2 (PAR-2) pathway. The importance of the interplay between genetic and environmental factors is evidenced by the enhancement of gluten immunopathology in a mouse model expressing HLA-DQ8 CeD risk genes. Thus, correction of bacterial proteolytic imbalance in the small intestine may constitute a new therapeutic target to prevent or ameliorate food sensitivities triggered by specific protein antigens.

## Results

### Characteristic microbial proteolytic phenotype in CeD.
We collected biopsies from the second portion of the duodenum (demographic characteristics outlined in Supplementary Table 1), the site most commonly affected in CeD, from a cohort of patients with and without CeD (controls) and determined the proteolytic activity against gluten ("glutenasic" activity). Biopsies from CeD patients produced a larger hydrolytic halo in solid gluten-containing media compared with biopsies from controls (Fig. 1a). In accordance with previous studies[20,21], analysis of mucosa-associated microbiota by 16S sequencing showed that Proteobacteria and Firmicutes were the most representative phylum in duodenal samples, followed by Bacteroidetes, Tenericutes, and Actinobacteria (Fig. 1b). The microbiota was specific

of each person at the genus level (Fig. 1c) and no major differences were found in alpha- or beta-diversity between groups (Supplementary Figure 1a and 1b). However, glutenasic activity inversely correlated with Firmicutes and Bacteroidetes and directly correlated with Proteobacteria relative abundance (Fig.1d). At the genus level, increased glutenasic activity correlated with an increase in Proteobacteria such as Pseudomonas and Janthinobacterium and a reduction in core members of the duodenal microbiota, such as Lactobacillus and Clostridium (Fig. 1e). These results suggest an association between the small intestinal glutenasic profile in patients with CeD and the relative abundance of certain duodenal microbial groups, notably gluten-degrading bacteria such as Pseudomonas[18,22].

### LasB induces microbial shifts and innate immunity.
Pathogenic bacteria possess an arsenal of virulence factors that allow them to survive in the host and cause disease. Among such virulence factors, secreted extracellular proteases facilitate bacterial colonization by inducing damage to host tissue and actively subverting immune responses[10]. Due to the correlation between glutenasic activity in duodenal biopsies and Pseudomonas, a group previously found to metabolize gluten through LasB elastase[18], we chose P. aeruginosa-producing LasB elastase as a model bacterium for in vivo analysis. P. aeruginosa strain PA14[18] and a P. aeruginosa PA14 mutant with a transposon disruption in the lasB gene[19] were used. We confirmed that the lasB mutant lacked elastase activity compared to the wild-type (WT) P. aeruginosa PA14 strain (Supplementary Figure 2a) in vitro. The functional capacity of LasB to degrade gluten was confirmed using a commercially available purified LasB elastase (Supplementary Figure 2b). As shown before[18], degradation of gluten occurs in the presence of supernatant extracted from P. aeruginosa PA14 WT, but not from P. aeruginosa lasB transposon mutant (Supplementary Figure 2c-left). Genetic complementation of the lasB mutant using a vector harboring the WT lasB gene under the control of its native promoter or an inducible arabinose promoter restored gluten degradation of the lasB mutant, while the empty vector did not restore activity (Supplementary Figure 2c-center). Of note, expression of lasB into Escherichia coli resulted in gluten degradation under the arabinose inducible promoter, but not when repressed by the addition of glucose (Supplementary Figure 2c-right). Altogether, these data demonstrate that LasB is responsible for the observed in vitro gluten degradation by P. aeruginosa.

C57BL/6 mice harboring a limited microbiota derived from the altered Schaedler flora (clean specific pathogen free (SPF)) were then supplemented with P. aeruginosa PA14 containing a lasB-lux transcriptional construct integrated on its genome[23] (Supplementary Table 2). This reporter strain allows direct monitoring of the lasB promoter activity via the detection of light (Fig. 2a, Supplementary Figure 2d). The experiment confirmed the presence of P. aeruginosa and the expression of lasB in the small intestine. In subsequent experiments, clean SPF C57BL/6 mice were colonized with P. aeruginosa PA14 WT, or a P. aeruginosa PA14 mutant with a transposon disruption in the lasB gene. Mice were then treated with gluten (gluten sensitization and challenge) or without gluten (sham; controls) (Fig. 2b)[24]. Small intestinal bacterial load was similar in P. aeruginosa PA14 WT and lasB mutant colonized mice, with an increase of P. aeruginosa counts in mice treated with gluten (Fig. 2c). Bacterial strains in colonized mice maintained their in vitro proteolytic phenotype (Supplementary Figure 2e) and mice colonized with P. aeruginosa PA14 WT had higher small intestinal elastolytic activity, but similar tryptic activity levels, compared with mice colonized with lasB mutant (Fig. 2d and Supplementary Figure 2f). This was

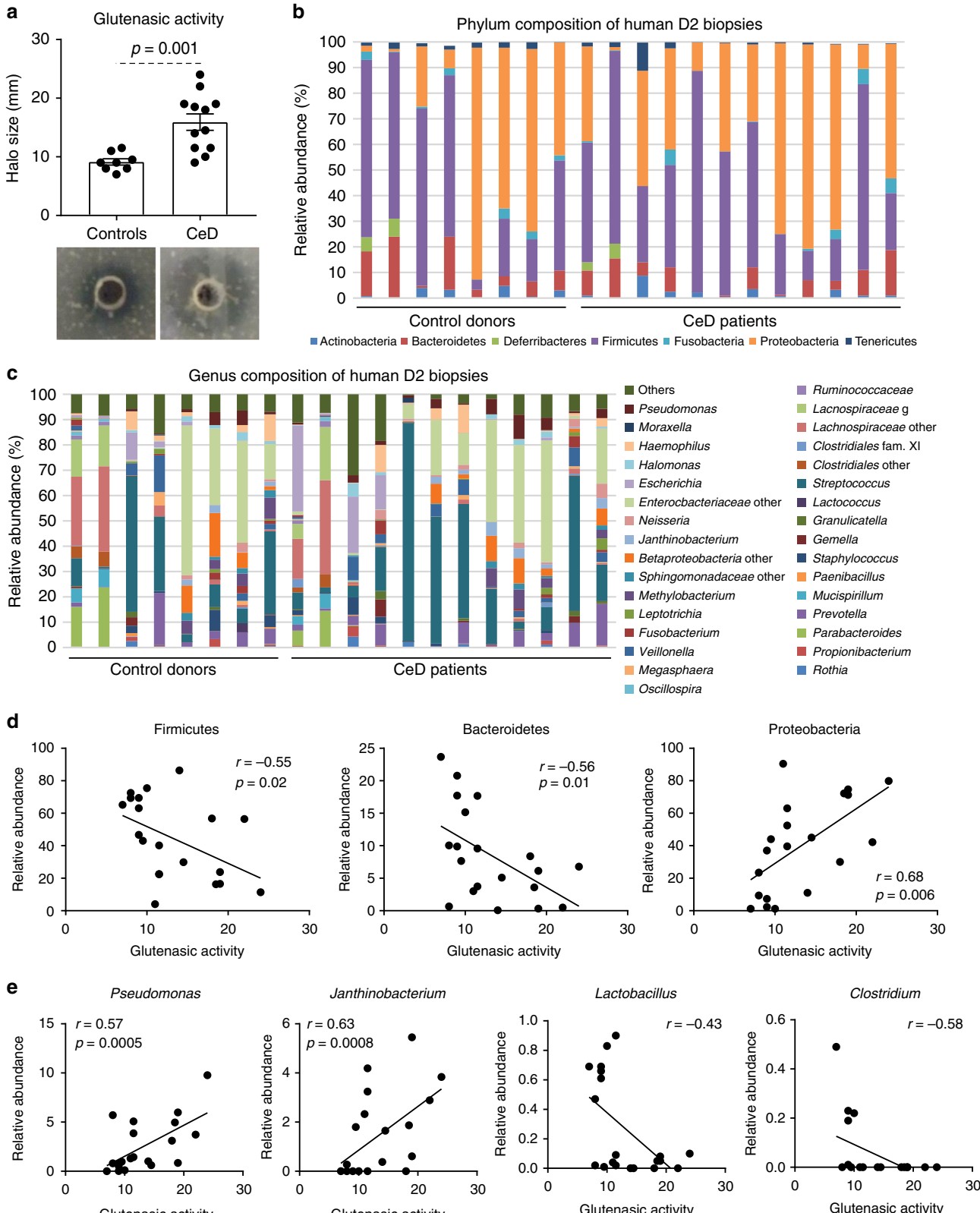

**Fig. 1** High proteolytic activity in celiac disease (CeD) duodenum correlates with microbiota changes. **a** Glutenasic activity measured in duodenal biopsies from non-celiac disease donors (controls; $n = 8$) and CeD patients ($n = 12$). Data presented as mean ± s.e.m. where each dot represents an individual human donor. Displayed $P$ value was calculated by Student's $t$-test. Representative bioassays are shown. **b**, **c** Relative abundance of the microbial composition at the phylum level (**b**) and genus level (**c**) of duodenal biopsies from controls ($n = 8$) and CeD patients ($n = 12$). **d**, **e** Correlation between microbial relative abundance at the phylum level (**d**) or genus level (**e**) and glutenasic activity in duodenal biopsies from controls ($n = 8$) and CeD ($n = 12$) human donors. Displayed $P$ values survived 10% false discovery rate (FDR) correction. Each dot represents individual human donor. Correlation based on Spearman's index

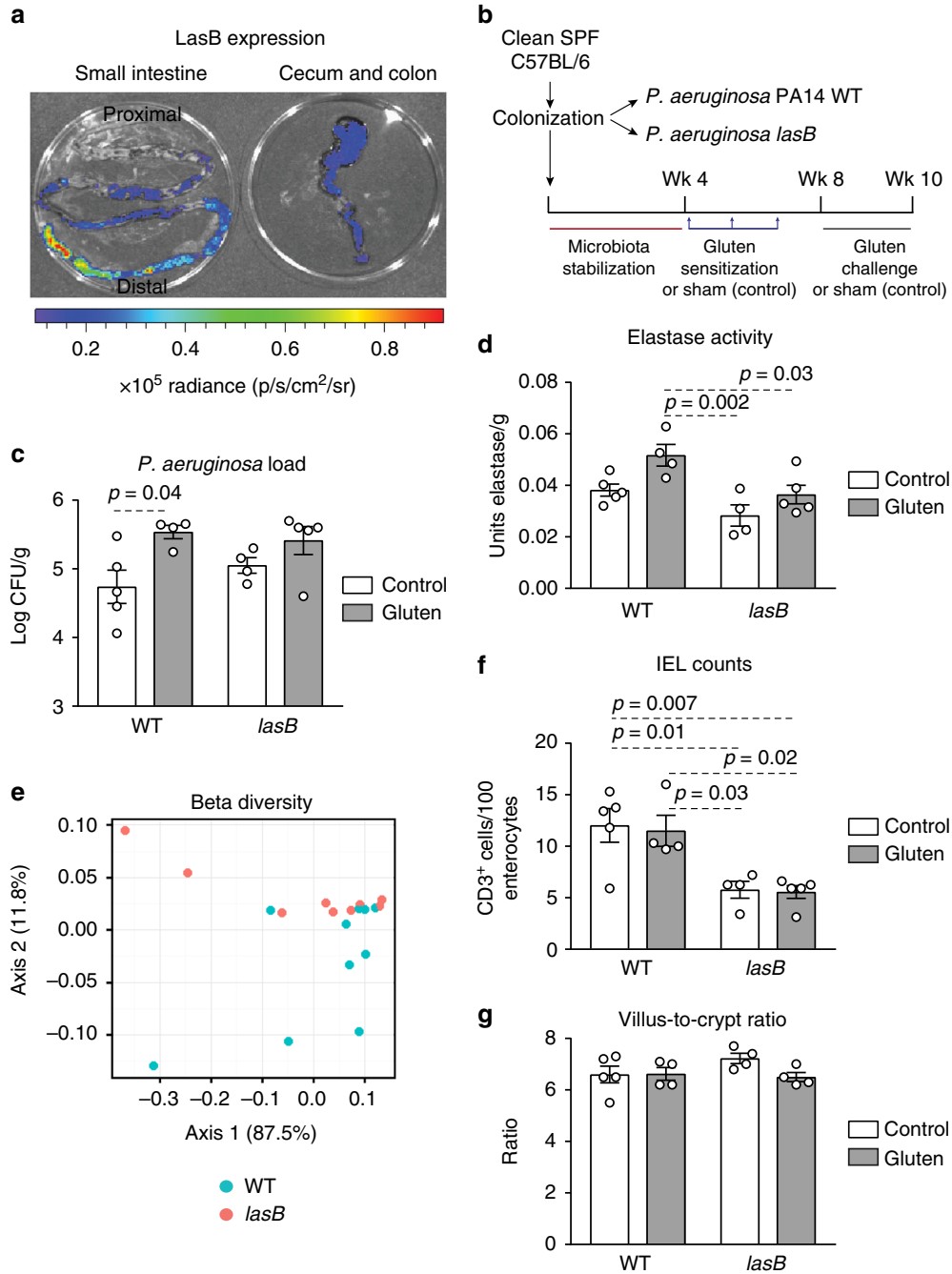

**Fig. 2** LasB induces intraepithelial lymphocytes (IELs) and microbiota shifts. **a** Visual expression of *lasB* in the small intestine (left panel) and large intestine (right panel) of clean specific pathogen free (SPF) mice 2 weeks following colonization with a luminescent producing *P. aeruginosa* mutant linked to the *lasB* promoter (*n* = 3). One representative image is shown. **b** Protocol for gluten sensitization and challenge (gluten treatment) in clean SPF C57BL/6 mice colonized with *P. aeruginosa* PA14 WT (wild-type (WT) gluten) or the *lasB* mutant (*lasB* gluten). Sham controls consisted of non-sensitized and non-gluten challenged clean SPF C57BL/6 mice colonized with *P. aeruginosa* PA14 WT (WT control) or the *lasB* mutant (*lasB* control). **c** *P. aeruginosa* bacterial load in the small intestine of clean SPF C57BL/6 colonized with *P. aeruginosa* PA14 WT or *lasB* mutant with gluten treatment (gray bars; *n* = 4 WT, *n* = 5 *lasB*) or without gluten treatment (white bars; *n* = 5 WT, *n* = 4 *lasB*). **d** Elastase activity measured in the small intestine of clean SPF C57BL/6 mice colonized with *P. aeruginosa* PA14 WT or the *lasB* mutant with gluten treatment (gray bars; *n* = 4 WT, *n* = 5 *lasB*) or without gluten treatment (white bars; *n* = 5 WT, *n* = 4 *lasB*). **e** β-Diversity Bray–Curtis dissimilarity principal coordinates analysis (PCoA) plot of microbiota profiles from clean SPF C57BL/6 mice colonized with *P. aeruginosa* PA14 WT (*n* = 10) or the *lasB* mutant (*n* = 9). Each dot represents an individual mouse. Differences between bacterial communities were tested by permutational multivariate analysis of variance (PERMANOVA) using Qiime. **f** Quantitative measurement of IELs/100 enterocytes in small intestinal villi tips of clean SPF C57BL/6 mice colonized with *P. aeruginosa* PA14 WT or the *lasB* mutant, with gluten treatment (gray bars; *n* = 4 WT, *n* = 5 *lasB*) or without gluten treatment (white bars; *n* = 5 WT, *n* = 4 *lasB*). **g** Small intestinal villus-to-crypt ratios of clean SPF C57BL/6 mice colonized with *P. aeruginosa* PA14 WT or the *lasB* mutant, with gluten treatment (gray bars; *n* = 4 WT, *n* = 5 *lasB*) or without gluten treatment (white bars; *n* = 5 WT, *n* = 4 *lasB*). Data for **c**, **d**, **f**, **g** are presented as mean ± s.e.m. where each dot represents an individual mouse. Displayed *P* values were calculated by a one-way analysis of variance (ANOVA) with Tukey's post-hoc test

associated with differences in small intestinal microbiota composition between the two groups (Fig. 2e and Supplementary Figure 2g). While *P. aeruginosa* PA14 WT, but not *lasB* mutant colonized mice, had increased small intestinal IEL (CD3-positive cells (CD3+)) counts (Fig. 2f), there were no differences in villus-to-crypt ratios (Fig. 2g), levels of anti-gliadin antibodies (Supplementary Figure 2h), or in the small intestinal expression of 254 inflammatory genes (Nanostring Mouse Inflammation Panel; raw data included in Expression Omnibus database (GEO, GSE125983) under the Umbrella project PRJNA518891). These data show that in C57BL/6 mice, LasB is associated with higher IEL counts and altered small intestinal microbiota profiles, in the absence of significant mucosal damage.

**LasB is pro-inflammatory in the absence of a microbiota.** To rule out possible contributions of shifts in clean SPF microbiota profiles induced by the presence of the opportunistic pathogen, germ-free C57BL/6 mice were monocolonized with *P. aeruginosa* PA14 WT, or its *lasB* mutant (Fig. 3a). Using the *lasB-lux* transcriptional reporter strain, we confirmed that the *lasB* promoter was transcriptionally active under these colonization conditions (Supplementary Figure 3a). On killing of mice, we recovered $10^3$–$10^4$ colony-forming units (cfu)/g of small intestinal content in each group (Fig. 3b). *P. aeruginosa* PA14 WT-monocolonized mice had higher glutenasic and elastolytic activity (Fig. 3c, d), but not tryptic activity (Supplementary Figure 3b) in small intestinal washes, as well as higher small intestinal IEL counts (Fig. 3e) compared with mice colonized with the *lasB* mutant. Flow cytometry confirmed that colonization with *P. aeruginosa* PA14 WT increased total numbers of IELs positive for CD45, CD3, and CD103 (Supplementary Figure 3c and 3d).

Small intestinal washes from mice monocolonized with *P. aeruginosa* PA14 WT also had higher mucolytic activity compared with washes from mice colonized with the *lasB* mutant (Supplementary Figure 3e). We thus investigated mucus invasion by *P. aeruginosa* using 16S-fluorescence in situ hybridization (FISH). *P. aeruginosa* PA14 WT encroached upon the epithelial layer with greater capacity than the *lasB* mutant, suggesting a higher invasive phenotype facilitated by LasB elastase (Supplementary Figure 3f). We did not find evidence of *Pseudomonas* translocation to the spleen (Supplementary Figure 3g), suggesting that these effects were limited to the mucosal surface.

No differences in messenger RNA expression of a panel of genes involved in immune pathways were found between groups, using whole small intestinal tissues (Nanostring Mouse Inflammation Panel; raw data included in GEO (GSE125983) under the Umbrella project PRJNA518891). Analysis using an isolated cell compartment enriched for IELs revealed that the transcription of 40 genes was significantly increased in mice monocolonized with *P. aeruginosa* PA14 WT compared to mice monocolonized with the *lasB* mutant. The functions of these genes related to IEL activation and cytotoxicity, barrier function, and resistance to infection (Fig. 3f and Supplementary Figure 4a). Using Ingenuity Pathway Analysis (IPA) software (Qiagene), overexpressed genes were found to be related to canonical pathways relevant to infection or recognition of bacteria (*ifng, il6, il17a, tnf,* and *il22*), and inflammation mediated by several pathways including G protein-coupled receptor signaling (*c3ar1, cysltr1, stat3*). Pathways were associated to CD3+ cells and cell death based on overexpression of genes such as *fasl, tgfb1, ltb, lta,* and *map3k9*. The changes in gene expression were also linked with adaptive immune responses mediated by major histocompatibility

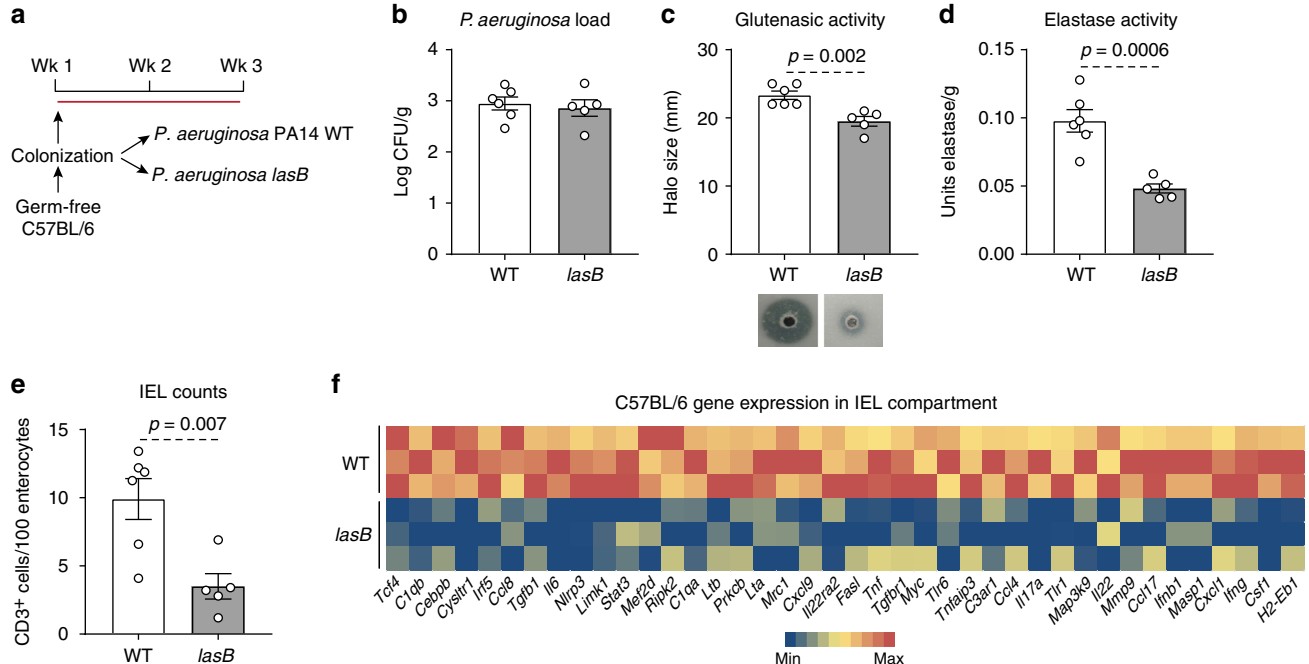

**Fig. 3** LasB induces a pro-inflammatory response in the absence of microbiota. **a** Protocol for monocolonization of C57BL/6 mice with *P. aeruginosa* PA14 wild-type (WT) or the *lasB* mutant. **b** *P. aeruginosa* bacterial load recovered in the small intestine of WT (*n* = 6) and *lasB* mutant (*n* = 5) colonized mice 3 weeks post colonization. **c** Luminal small intestinal glutenasic activity from mice monocolonized with *P. aeruginosa* PA14 WT (*n* = 6) or the *lasB* mutant (*n* = 5). Representative bioassays are shown. **d** Luminal small intestinal elastase activity from mice monocolonized with *P. aeruginosa* PA14 WT (*n* = 6) or the *lasB* mutant (*n* = 5). **e** Quantitative measurement of intraepithelial lymphocytes (IELs)/100 enterocytes in small intestinal villi tips of *P. aeruginosa* PA14 WT (*n* = 6) or *lasB* mutant (*n* = 5) monocolonized mice. **f** Heat map of gene expression in the small intestinal IEL compartment of mice colonized with *P. aeruginosa* PA14 WT (*n* = 3) or the *lasB* mutant (*n* = 3) assessed by NanoString nCounter gene expression. Only statistically different genes generated by nSolver 2.5 using Student's *t*-test and based on bacterial colonization are shown. Data for **b**–**e** are presented as mean ± s.e.m. where each dot represents an individual mouse. Displayed *P* values were calculated by Student's *t*-test

complex (MHC) class II genes and immunoglobulin production (Supplementary Figure 4b). Thus, our experimental data suggest that LasB-producing *P. aeruginosa* is sufficient to promote a widespread pro-inflammatory molecular signature in the IEL compartment of the duodenum.

**Cleavage of PAR-2 by LasB mediates IEL increase.** Protease-activated receptors (PARs) are a subfamily of G protein-coupled receptors that are activated by the proteolytic cleavage of part of their extracellular domain[25,26]. Using a transgenic cell line harboring PAR-2 tagged to luciferase[27], we found that

*P. aeruginosa* PA14 WT, but not the *lasB* mutant, cleaved the external domain of PAR-2, suggesting that bacterial elastase is responsible for the activation of this pathway (Fig. 4a). We confirmed that genetic complementation of the *P. aeruginosa lasB* transposon mutant with the *lasB*-expressing plasmid restored the capacity of the strain to degrade the external domain of PAR-2 (Supplementary Figure 5a). In addition, transformation of an *E. coli* strain (with no capacity to degrade PAR-2) with the plasmid coding for *lasB* resulted in high degradation of the PAR-2 receptor under the arabinose inducible promoter and much lower degradation when repressed by the addition of

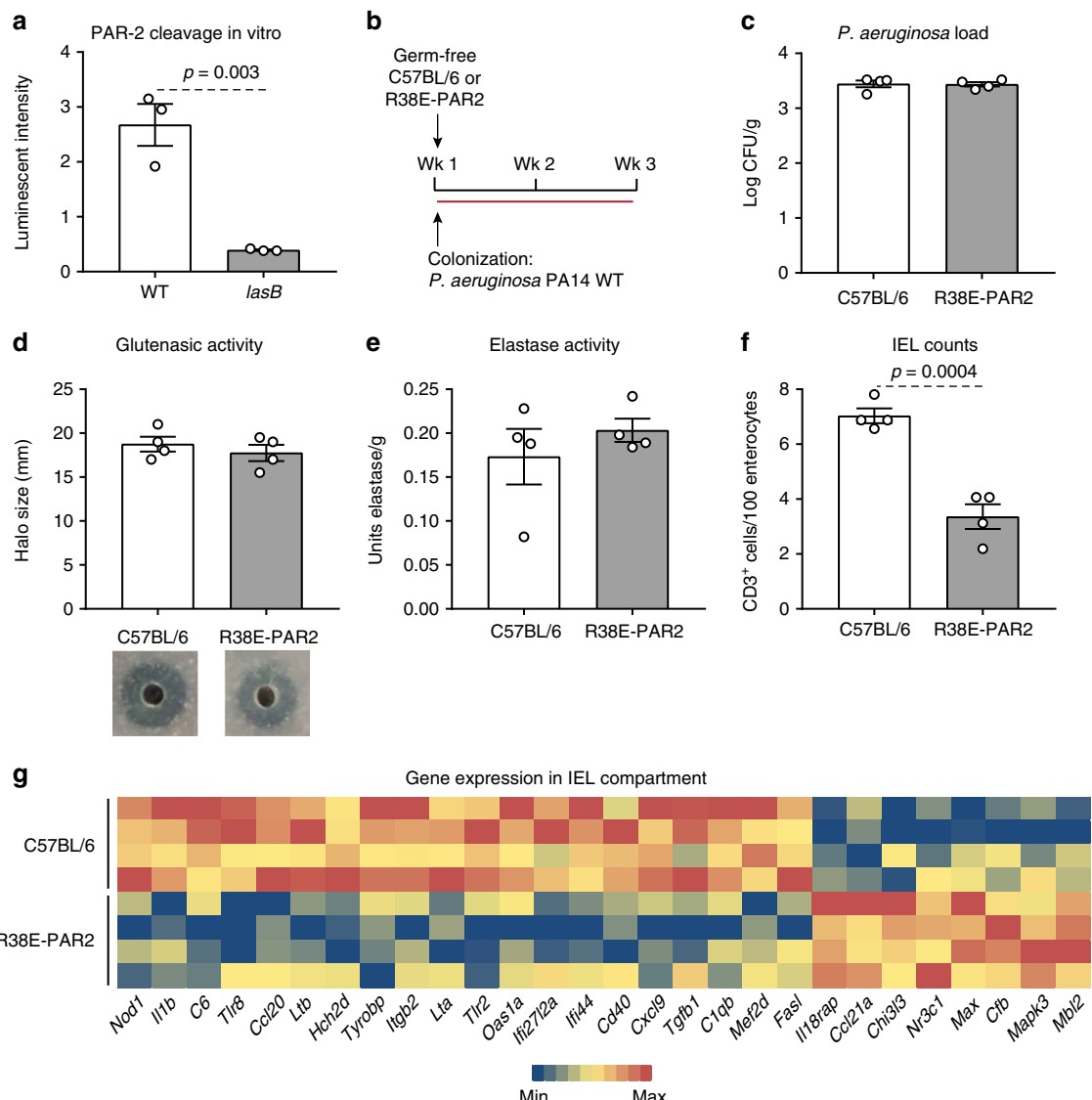

**Fig. 4** LasB induces a pro-inflammatory response through protease-activated receptor-2 (PAR-2). **a** In vitro cleavage of the external domain of PAR-2 by *P. aeruginosa* PA14 wild-type (WT) (*n* = 3 biological replicates) and the *lasB* mutant (*n* = 3 biological replicates). Data presented as mean ± s.e.m where each dot represents one biological replicate. **b** Protocol for monocolonization of germ-free C57BL/6 mice and protease-resistant PAR-2 mutant mice (PAR38E-PAR2) with *P. aeruginosa* PA14 WT. **c** *P. aeruginosa* bacterial load recovered in the small intestine of monocolonized C57BL/6 mice (*n* = 4) and PAR38E-PAR2 mice (*n* = 4). **d** Luminal small intestinal glutenasic activity from C57BL/6 mice (*n* = 4) and PAR38E-PAR2 mice (*n* = 4) monocolonized with *P. aeruginosa* PA14 WT. Representative bioassays are shown. **e** Luminal small intestinal elastase activity from C57BL/6 mice (*n* = 4) and PAR38E-PAR2 mice (*n* = 4) monocolonized with *P. aeruginosa* PA14 WT. **f** Quantitative measure of intraepithelial lymphocytes (IELs)/100 enterocytes in small intestinal villi tips of C57BL/6 mice (*n* = 4) and R38E-PAR2 mice (*n* = 4) monocolonized with *P. aeruginosa* PA14 WT. **g** Heat map of gene expression in the IEL compartment of the small intestine of C57BL/6 mice (*n* = 4) and PAR38E-PAR2 mice (*n* = 4) monocolonized with *P. aeruginosa* PA14 WT assessed by NanoString nCounter gene expression. Only statistically different genes generated by nSolver 2.5 using Student's *t*-test and based on bacterial colonization are shown. Data for **c–e** are presented as mean ± s.e.m. where each dot represents an individual mouse. Displayed *P* values were calculated by Student's *t*-test

glucose (Supplementary Figure 5a). These results demonstrate the capacity of LasB to cleave the external domain of PAR-2.

To test the role of PAR-2 in the immune response mediated by LasB in vivo, mice monocolonized with *P. aeruginosa* PA14 WT were intraperitoneally injected every 3 days with a PAR-2 antagonist or vehicle for 2 weeks (Supplementary Figure 5b). Both groups showed similar bacterial load and glutenasic activities in the small intestine (Supplementary Figure 5c and 5d). However, mice monocolonized with *P. aeruginosa* PA14 WT and treated with the GB83 PAR-2 antagonist had lower IEL counts compared to mice treated with vehicle (Supplementary Figure 5e), suggesting a role for PAR-2.

Protease-resistant PAR-2 mutant (R38E-PAR2) mice were next used to determine the role of the external domain of PAR-2 in LasB-mediated innate immune responses (Supplementary Figure 5f). Supplementation of *P. aeruginosa* PA14 WT to SPF C57BL/6 mice, but not to SPF R38E-PAR2 mice, led to higher small intestinal IEL counts, despite both groups harboring similar luminal small intestinal elastase activity (Supplementary Figure 5g and 5h). To rule out other microbial contributions, germ-free R38E-PAR2 and C57BL/6 mice were then monocolonized with *P. aeruginosa* PA14 WT (Fig. 4b). We recovered similar counts of *P. aeruginosa* in the small intestine, and there was no difference in glutenasic or elastase activity between groups (Fig. 4c–e). However, monocolonization increased small intestinal IEL counts only in C57BL/6 mice, suggesting PAR-2 cleavage of the external domain mediates the innate activation mediated by LasB (Fig. 4f). Gene expression analysis in the isolated cell compartment enriched for IELs revealed that monocolonization led to overexpression of a total of 20 inflammatory genes, and underexpression of 8 genes in C57BL/6 mice compared with R38E-PAR2 mice (Fig. 4g). Specifically, genes related to recognition of bacteria (*tlr*2, *tlr*8, and *nod*1) and apoptosis (*tgfb*1, *lta*, and *fasl*) were increased in C57BL/6 mice, while genes such as *Il18rap* and *Mapk*3 were increased in R38E-PAR2 mice. Expression of genes specifically increased by *P. aeruginosa* WT in C57BL/6 mice (Fig. 3f) related to CD3+ cells and cell death (*tgfb*1, *lta*, *ltb*, and *fasl*) were underexpressed in R38E-PAR2 mice monocolonized with *P. aeruginosa* WT (Fig. 4g and Supplementary Figure 5i). These results suggest that proteolytic cleavage of PAR-2 receptor is key for the IEL responses induced by bacterial elastase.

**LasB enhances gluten-induced pathology**. We used an animal model of gluten sensitivity to investigate the effects of bacterial proteases and innate immune activation in a genetically susceptible host. We have previously shown that responses to gluten in mice expressing the HLA-DQ8 susceptibility gene are microbiota dependent, with unclear mechanisms. Clean SPF NOD/DQ8 mice are protected from developing immunopathology, whereas mice that harbor a more diverse and complex intestinal microbiota that includes Proteobacteria develop moderate immunopathology when sensitized with gluten[28]. Therefore, we colonized clean SPF NOD/DQ8 mice with *P. aeruginosa* PA14 WT or the *lasB* mutant, and treated them with or without gluten (Fig. 5a). *P. aeruginosa* successfully colonized the small intestine, where higher elastase activity was observed in *P. aeruginosa* PA14 WT compared with *lasB* mutant colonized mice (Fig. 5b, c). No differences in tryptic activity were observed between groups (Supplementary Figure 6a). Differences in clean SPF microbiota profiles (Supplementary Figure 6b and 6c) between the two groups of mice were also detected. Compared to mice colonized with the lasB mutant, *P. aeruginosa* PA14 WT-colonized mice had higher small intestinal IEL counts (Fig. 5d), increased PAR-2 expression in the small intestine epithelium (Supplementary Figure 6d), and increased inflammatory gene expression in whole small intestinal sections,

which were independent of gluten treatment. Genes related to resistance to infection, cytotoxicity, cell death, and G protein-coupled receptor signaling such as *c3ar1*, *hif1α*, *il22*, and *ifnγ* were highly expressed in mice colonized with *P. aeruginosa* PA14 WT (Fig. 5e and Supplementary Figure 6e). Gluten treatment was associated with higher expression of additional inflammatory genes such as *il18* (Fig. 5f and Supplementary Figure 6f). Moreover, the presence of *P. aeruginosa* PA14 WT resulted in lower small intestinal villus-to-crypt ratios (Fig. 5g), and higher anti-gliadin antibody titers in small intestinal washes, only after gluten treatment (Supplementary Figure 6g). Thus, in a mouse model with genetic susceptibility, LasB enhances gluten immunopathology through gluten-independent mechanisms.

**CeD microbiota increases proteolytic activity and IELs**. The previous experiments determined a critical role of bacterial glutenasic activity in host immune responses to dietary protein antigen; however, its role in the presence of a bacterial community derived from the human gut is unclear. To address this, we colonized germ-free C57BL/6 mice using duodenal aspirates from 9 randomly selected donors with active CeD (n = 4) or without CeD (controls; n = 5) for 3 weeks (Fig. 6a). Duodenal aspirates come from the same cohort as Fig. 1. Compared with fecal slurries, duodenal aspirates have a low bacterial yield, thus each donor allowed for the colonization of 2–3 mice after adjusting for equal bacterial concentration in the inoculum. Small intestinal microbiota profiles in recipient mice clustered significantly by donor (Fig. 6b). All the bacteria found in the small intestine of the recipient mice (>1% abundance) were present in the donors and between 20 and 80% of the total human microbiota composition was transferred into the mice (Supplementary Figure 7a and 7b). No differences in small intestinal microbiota composition (Supplementary Figure 7a) or diversity (Fig. 6b) were found in recipient mice based on CeD diagnosis. Although individual donor-related differences were observed, pooled group data revealed that mice colonized with microbiota from CeD had higher small intestinal glutenasic activity (Fig. 6c) and higher small intestinal IEL counts (Fig. 6d). Moreover, there was a significant correlation between glutenasic activity and IEL counts (Fig. 6e). As in human duodenal biopsies, glutenasic activity inversely correlated with Firmicutes and directly correlated with Proteobacteria relative abundance in the small intestine of recipient mice (Fig. 6f). Taken together, these results indicate that the increased small intestinal microbial glutenasic activity in CeD patients can be transferred to germ-free mice where it associates with higher IEL counts.

**Discussion**

Approaches to prevent or treat food sensitivities require knowledge of the complex mechanisms and signaling events by which environmental factors interact with food triggers to produce a response in the host[29–31]. Here, we demonstrate that duodenal biopsies from patients with active CeD, a condition that develops when oral tolerance to dietary gluten is broken, have increased proteolytic activity against gluten peptides compared with healthy controls that correlates with Proteobacteria abundance. Several clinical and cohort studies have linked intestinal infections and changes in the intestinal microbiota, including an increase in Proteobacteria, with CeD onset or activity[15]. These microbial changes in CeD could represent a disease-triggering or -modifying factor, or they could be a consequence of disease driven by intestinal damage or malabsorption. However, knowledge of specific mechanisms by which the small intestinal microbiota influence CeD pathogenesis is lacking.

The ubiquitous presence of gluten in the diet, and the resistance of gluten proteins to gastrointestinal digestion[11], allow

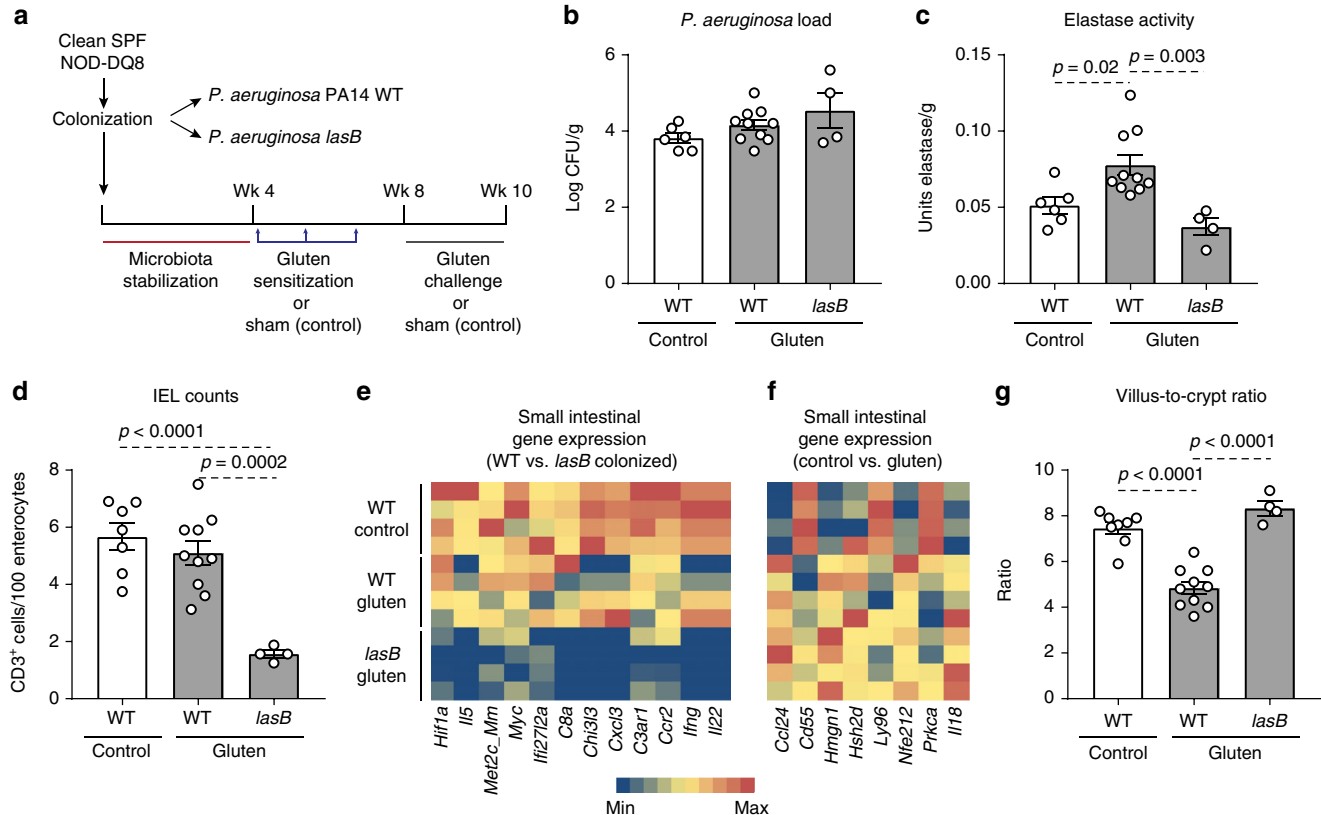

**Fig. 5** LasB enhances gluten sensitivity in genetically predisposed mice. **a** Protocol for gluten sensitization and challenge (gluten treatment) of clean specific pathogen free (SPF) NOD/DQ8 mice colonized with *P. aeruginosa* PA14 WT (wild-type (WT) gluten) or the *lasB* mutant (*lasB* gluten). Sham controls consisted of non-sensitized and non-gluten challenged clean SPF NOD/DQ8 mice colonized with *P. aeruginosa* PA14 WT (WT control). **b** *P. aeruginosa* bacterial load in the small intestine of clean SPF NOD/DQ8 colonized with *P. aeruginosa* PA14 WT or the *lasB* mutant, treated with gluten (gray bars; $n = 10$ WT, $n = 4$ *lasB*) or without gluten (white bars; $n = 6$). **c** Luminal small intestinal elastase activity of clean SPF NOD/DQ8 mice colonized with *P. aeruginosa* PA14 WT or the *lasB* mutant, treated with gluten (gray bars; $n = 10$ WT, $n = 4$ *lasB*) or without gluten (white bars; $n = 6$). **d** Quantitative measure of intraepithelial lymphocytes (IELs)/100 enterocytes in small intestinal villi tips in clean SPF NOD/DQ8 mice colonized with *P. aeruginosa* PA14 WT or the *lasB* mutant, treated with gluten (gray bars; $n = 10$ WT, $n = 4$ *lasB*) or without gluten (white bars; $n = 7$). **e, f** Heat map of significantly altered genes in whole small intestinal tissue of clean SPF NOD/DQ8 mice ($n = 4$/group), when comparisons were performed between *P. aeruginosa* PA14 WT and *lasB* mutant colonized mice (**e**) or when comparisons were performed between gluten treated and control mice (**f**). Gene expression was assessed by NanoString nCounter gene expression and only significantly altered genes generated by nSolver 2.5 using multiple *t*-tests are shown. **g** Small intestinal villus-to-crypt ratios in clean SPF NOD/DQ8 mice colonized with *P. aeruginosa* PA14 WT or the *lasB* mutant, treated with gluten (gray bars; $n = 10$ WT, $n = 4$ *lasB*) or without gluten (white bars; $n = 8$). Data for **b**–**d**, **g** are presented as mean ± s.e.m. where each dot represents an individual mouse. Displayed *P* values were calculated by one-way analysis of variance (ANOVA) with Tukey's post-hoc test

for the presence of long immunogenic peptides in the gut lumen that are substrates for microbial metabolism[32,33]. Proliferation of specific bacterial groups with the ability to metabolize gluten could promote sensitivity in a predisposed host. Opportunistic pathogens, including members of Proteobacteria such as *P. aeruginosa* isolated from the duodenum of CeD patients, can metabolize gluten predigested by human proteases into shorter immunogenic peptides that permeate better through the barrier and stimulate human gluten-specific T cells[18]. However, it is unknown whether the proteases produced by these opportunistic pathogens can independently disrupt host inflammatory pathways relevant for the generation of sensitivity to dietary protein antigens. Here, we found a correlation between *Pseudomonas* relative abundance and increased proteolytic activity against gluten in the small intestine of patients with CeD, which can be transferred to gnotobiotic mouse models.

We first performed a series of gnotobiotic experiments using clean SPF C57BL/6 mice co-colonized with *P. aeruginosa* PA14 WT or a mutant lacking *lasB* activity. We detected different small

intestinal microbiota profiles between the two groups, suggesting the presence of bacterial protease could shift the resident microbiota. This is consistent with clinical studies describing a high abundance of opportunistic pathogens in the microbiota of patients with active CeD[34]. However, monocolonization of germ-free mice with *P. aeruginosa* PA14 WT, or the *lasB* mutant, indicated a protease–host interaction was sufficient to induce the inflammatory phenotype. *P. aeruginosa*-producing elastase led to higher IELs and overexpression of several pro-inflammatory genes in the IELs isolated from the small intestinal mucosa, including *tgfb* and *tnf*, two cytokines involved in IEL activity[35,36]. The expression of *ifng*, a cytokine central to the pathogenesis of CeD[37], and *fasl*, a mediator of IEL cytotoxicity[38], were upregulated in the compartment enriched for IELs. A number of genes involved in resistance to infection such as *il17*, *il22*, and *il6* were also increased[39,40]. The overexpression of these genes suggests a response related to infection, as well as activation of an innate immune response (*c3ar*, *cysltr1*, *stat3*), that could lead to cell death mediated by overexpression of *fasl*. This canonical pathway

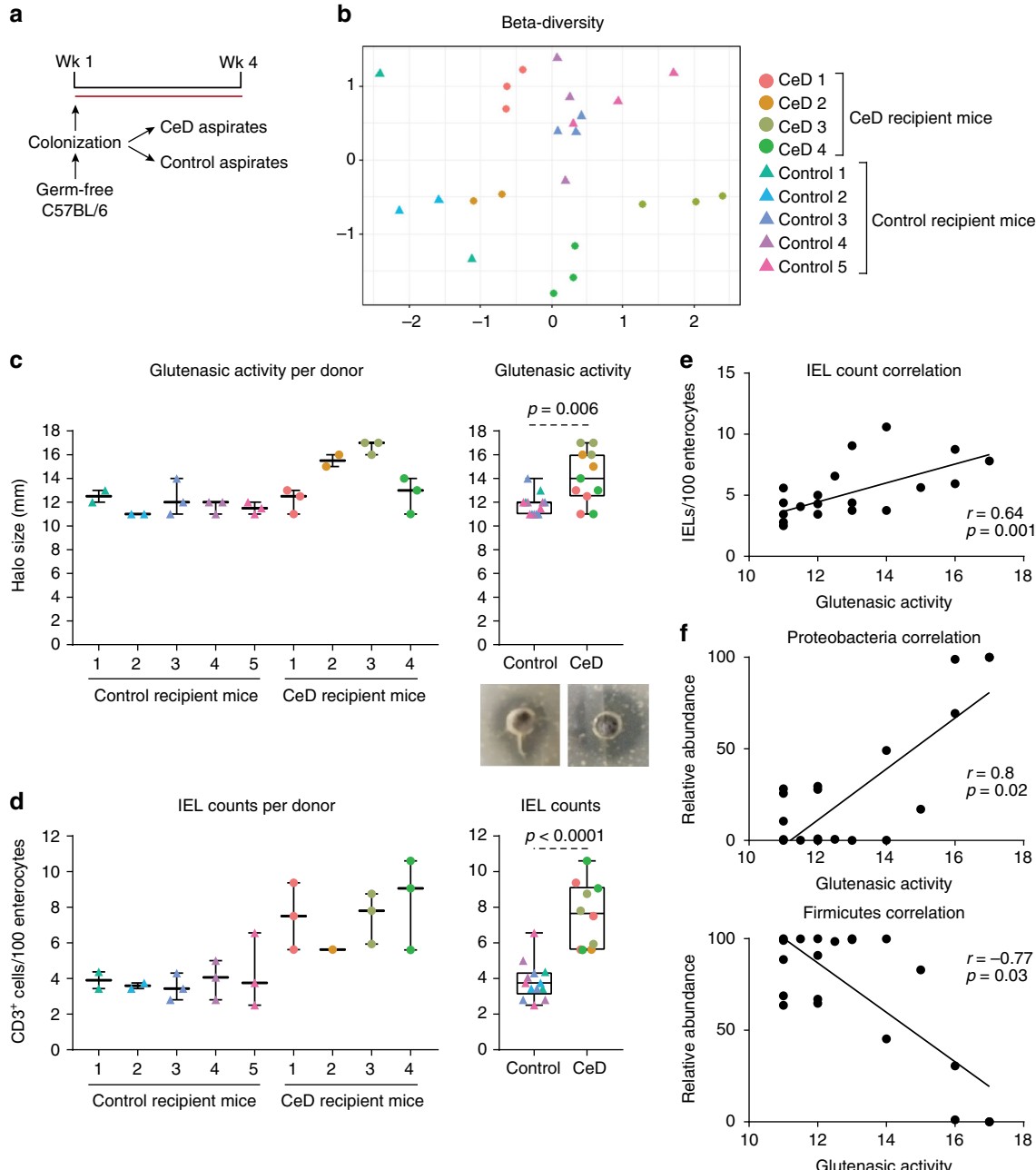

**Fig. 6** Celiac duodenal microbiota induces proteolytic activity and intraepithelial lymphocytes (IELs). **a** Protocol for the colonization of germ-free C57BL/6 mice with human small intestinal aspirates from celiac disease (CeD) patients (n = 4) or controls (n = 5). **b** Beta-diversity of small intestinal microbiota profiles of recipient mice colonized with human aspirates of patients with CeD and without CeD (controls), using Bray–Curtis dissimilarity represented as non-metric multidimensional scaling (NMSD). Differences between bacterial communities were tested by permutational multivariate analysis of variance (PERMANOVA) using Qiime. Each donor was used to colonize 2–3 mice, and each dot represents an individual mouse. **c** Glutenasic activity per donor (left side) and pooled (right side), measured in the small intestine of mice colonized with small intestinal aspirates from control or CeD donors. Representative bioassays are shown. **d** Quantitative measurement of IELs/100 enterocytes in small intestinal villi tips of mice colonized with small intestinal aspirates from control or CeD donors. IEL counts per donor (left panel) and pooled counts (right panel) are shown. Data for **c**, **d** are presented as median with interquartile range and whiskers extending from minimum to maximum. Each dot represents an individual mouse. Each donor, as indicated by a different color, was used to colonize 2–3 mice. Displayed P values were calculated by Mann–Whitney test. Tissue from one recipient mouse receiving CeD donor 2 presented technical difficulties during embedding and processing and was dropped out. **e** Correlation between glutenasic activity and IEL counts in the small intestine of mice colonized with small intestinal aspirates (n = 23). Each dot represents an individual mouse. Correlation based on Spearman's index. **f** Correlation between relative abundance of Proteobacteria and Firmicutes and small intestinal glutenasic activity of mice colonized with small intestinal aspirates (n = 23). Each dot represents an individual mouse. Displayed P values from Proteobacteria and Firmicutes survived 10% false discovery rate (FDR) correction. Correlation based on Spearman's index

is also linked to CD3+ IEL increase and adaptive immune responses mediated by MHC class II genes and immunoglobulin production, which are characteristic of CeD[36]. However, despite the induction of a pro-inflammatory molecular signature within the IEL compartment of C57BL/6 mice colonized with *P. aeruginosa* PA14 WT, histological assessment did not reveal villus atrophy.

Mice monocolonized with *P. aeruginosa* PA14 WT had a higher presence of bacteria within the mucus layer, suggesting that bacterial elastase is also used for mucus invasion. This encroachment toward the mucosal surface was not associated with bacterial translocation to secondary lymphoid organs, and thus supports a physical mechanism through which *P. aeruginosa* elastase could facilitate contact with host epithelial cells to promote a localized pro-inflammatory response. As a molecular mechanism, we investigated the role of PARs, G protein-coupled receptors that are activated by limited proteolysis of their extracellular N-terminal domain, and are ubiquitously expressed in the gut. Activation of PARs induces a wide array of pro-inflammatory and proliferative effects[1,25,26]. Our study reveals that *P. aeruginosa* LasB cleaves the external domain of PAR-2 in vitro, suggesting activation of the receptor. This cleavage was also noted with an elastase-producing celiac clinical isolate (*P. aeruginosa* X46.1[22]), which also induced an increase of IELs in clean SPF C57BL/6 mice (Supplementary figure 8). Thus, the phenotype described is not restricted to *P. aeruginosa* PA14. *P. aeruginosa* has previously been shown to activate or disable PAR-2[41,42]. Specifically, Dulon et al.[41] demonstrated that LasB cleaves the PAR-2 receptor thereby altering the host innate defense mechanisms. Here, administration of *P. aeruginosa* PA14 WT to protease-resistant PAR-2 mutant mice did not increase IEL counts, implicating PAR-2 signaling, and specifically external domain cleavage by LasB (canonical or alternative activation), in the IEL response. It is also possible that other PARs such as PAR-1, -2, -3, and -4 could participate in the *P. aeruginosa* LasB-triggered events. There are several examples for biologically relevant PAR-1-PAR-2[43,44] and PAR-3-PAR-2 cross-activation[45]. However, it is important to mention that the protease-resistant PAR-2 mutant mice still respond to cross-activation by other PARs[46], strongly supporting that PAR-2 cleavage, rather than other receptor crosstalk, directly mediates the observed effects. Gene expression analysis revealed a reduction in pro-inflammatory genes mediating bacteria recognition (*tlr*2, *tlr*8, and *nod*1) in R38E-PAR2 mice colonized with *P. aeruginosa* PA14 WT, consistent with several studies linking PARs with the normal signaling of pathogen-associated molecular patterns[46–48]. In addition, genes related to apoptosis and cell death, including those specifically overexpressed in C57BL/6 mice monocolonized with *P. aeruginosa* PA14 WT (*tgfb*1, *lta*, *ltb* and *fasl*), were reduced in R38E-PAR2 mice. Some of these genes mediate CD3+ IEL increase (*tgfb*1) and their pathogenic capacity (*fasl*), which are characteristic in active CeD[38,49–51]. Thus, we demonstrate a role of microbial elastase in the recruitment of IELs in the small intestine. Although this mechanism has no major pathological consequences in C57BL/6 mice, it could contribute to villus damage in a genetically predisposed host.

Mice expressing the HLA-DQ8 gene in NOD background (NOD/DQ8) do not develop gluten immunopathology if they are reared in clean SPF conditions that lack pathobionts[28]. Here we found that colonization of clean SPF NOD/DQ8 mice with *P. aeruginosa* PA14 WT led to an innate immune activation and reduced villus-to-crypt ratios after gluten exposure. Thus, genetically susceptible mice protected by the presence of a innocuous microbiota develop moderate villus blunting when supplemented with elastase-producing *P. aeruginosa* and gluten, but not when supplemented with the same strain lacking elastase

activity. Gene expression analysis indicated that *P. aeruginosa* PA14 WT induced a pro-inflammatory cascade in whole small intestinal tissues of NOD/DQ8 mice involving *ifng* and *il22*. Genes related to cell death such as *hif1a* or *fasl*, which have previously been linked to CeD[38,52], were also highly expressed in NOD/DQ8 mice colonized with *P. aeruginosa* PA14 WT. The results suggest that elastase activity in *P. aeruginosa* PA14 WT drives pathways conducive to apoptosis in NOD/DQ8 mice, and illustrate the complexity of interactions between the dietary triggers, genetic predisposition, and specific microbial-induced pathways.

Microbiota-modulating therapies are currently employed empirically in clinical practice with little pathophysiological understanding. Given the increasing prevalence of adverse reactions to gluten and the recognition of the gluten-free diet as an imperfect therapy, there is interest in the development of adjuvant therapies to dietary restriction[53]. Our results identify an opportunistic pathogen-driven mechanism, through protease production and PAR-2 signaling, that could be exploited therapeutically to treat gluten-related disorders and other food sensitivities. Although animal models do not perfectly mimic human disease, models with a permissive genetic risk colonized with bacteria isolated from the human gut can be exploited to decipher pathways of interest that will inform clinical research[28]. The current results are consistent with previous studies showing dysregulation of eukaryotic protease inhibitors, such as elafin, in the duodenum of CeD patients[54], and the improvement of gluten sensitivity in mouse models after using bacteria producing specific protease inhibitors[55]. Future studies should investigate the modulation of this specific pathway in patients with CeD non-responsive to the gluten-free diet or in the prevention of other food sensitivities triggered by protein antigens.

## Methods

**Human samples**. We recruited a cohort of adult celiac patients ($n = 12$) attending the McMaster University Celiac Clinic (http://farncombe.mcmaster.ca/celiac-disease-clinic/) who were scheduled for upper gastrointestinal endoscopy. CeD patients showed clinical symptoms of disease, positive celiac serology markers (anti-tissue transglutaminase antibodies, DQ2/DQ8 genes), and a degree of duodenal mucosal atrophy, classified as type 3 according to the Marsh classification. Non-CeD controls ($n = 8$) were patients who attended the McMaster University Digestive Diseases Clinic for investigation of anemia, abdominal pain, or gastroesophageal reflux disease, and had normal endoscopy, with negative celiac serology, negative *Helicobacter pylori* test, and a normal villus structure by biopsy examination. Other organic diseases such as inflammatory bowel disease were also ruled out. CeD patients and controls were sex and age matched (Supplementary Table 1). None of the adults included in the study had been treated with antibiotics or proton-pump inhibitors for at least 1 month prior to the sampling time. All subjects signed the written informed consent and the study was approved by the Hamilton Integrated Research Ethics Board (REB # 12–599). Ethical regulations for work with humans and human samples were meticulously followed in the study. Biopsies from the second portion of the duodenum and aspirates from the small intestine were collected in all participants. The samples were frozen and stored until analysis.

**Mice**. Female and male 8- to 12-week-old germ-free C57BL/6 mice were generated by two-stage embryo transfer and bred under germ-free conditions in the Axenic Gnotobiotic Unit (AGU) at McMaster University. SPF C57BL/6 mice were originally purchased from Taconic and subsequently bred at McMaster's Central Animal Facility (CAF). Germ-free female and male 8 to 12-week-old Non-Obese Diabetic HLA-DQ8 (NOD/DQ8) transgenic mice[24,56] were generated by two-stage embryo transfer and bred in the AGU at McMaster University. Protease-resistant PAR-2 mutant breeding pairs (R38E-PAR2)[46] were originally provided by Wolfram Ruf from Johannes Gutenberg University Mainz and bred at McMaster's CAF under SPF conditions. R38E-PAR2 mice were re-derived germ-free by two-stage embryo transfer and bred under germ-free conditions in the AGU at McMaster University. Clean SPF mice originated from germ-free mice that were naturally colonized by co-housing with female mouse colonizers harboring altered Schaedler flora and bred for three generations in individually ventilated cage racks[28]. All mouse colonies were fed an autoclaved gluten-free mouse diet (Harlan, Indianapolis, IN) for two generations (Harlan Laboratories, Indianapolis, IN) until used in experiments. All mice had unlimited access to food and water. All experiments

were conducted with approval from the McMaster University Animal Care Committee and McMaster Animal Research Ethics Board (AREB) in an amendment to the Animal Utilization Protocol (AUP#170836). Ethical regulations for animal testing and research were followed in the study. Mice were sex and age matched.

**Bacterial strains and growth conditions.** We used the laboratory wild-type *P. aeruginosa* PA14 (burn patient isolate) for mouse experiments[19]. *P. aeruginosa lasB* mutant strain (*lasB::MAR2xT7*) from the available non-redundant transposon mutant library in strain PA14 was also used[19]. These strains were grown in Tryptic soy broth with gentamicin (50 μg/ml) to select for the Tn5 transposon. *P. aeruginosa* X46.1 isolated from the duodenum of a celiac patient was used in some experiments[22]. Proteolytic activities and phenotypes of all strains were monitored using gluten media. To measure expression from the *lasB* promoter in vivo, a *plasB-luxCDABE* transcriptional reporter[23] was integrated at the CTX integration site in the *P. aeruginosa* PA14 strain. Tetracycline at a concentration of 150 μg/ml was used to maintain the plasmids in *P. aeruginosa* strains, while 5 μg/ml was used for *E. coli* strains. Chemically competent NEB 5-alpha competent *E. coli* strain, derivate of the *E. coli* DH5α strain, and *E. coli* HB101 strain were used for plasmid construct transformation (New England Biolabs, Massachusetts, USA). *E. coli* strains were grown in Luria Bertani (LB) broth at 37 °C. All bacterial strains and vectors used in the study are described in Supplementary Table 2.

**Cloning and complementation of *lasB* transposon mutant.** A 1.8 bp DNA fragment containing the *lasB* gene, the endogenous terminator, and 300 bp of the upstream region were PCR amplified from *P. aeruginosa* PA14 using Q5 high fidelity polymerase (NEB, Massachusetts, USA) with the primers lasBF-end 5′-GTCGACTCTAGAGGATCCCCTGGCCCCTCGCTGAGCGC-3′ and lasBR-end 5′- AGAATTCGAGCTCGGTACCCCTGGCGGAAGACGGCGCTTGAGC-3′. A second fragment containing the *lasB* gene was also PCR amplified from *P. aeruginosa* PA14 using lasBF-ara 5′-AGAATTCGAGCTCGGTACCCCAGGAGAAC TGAACAAGATGAAGAAGG-3′ and lasBR-ara 5′-GTCGACTCTAGAGGATCC CCTTACAACGCGCTCGGGCA-3′. Then, both fragments were cloned into the commercial plasmid pHERD26T using Gibson cloning (New England). Briefly, the pHERD26T vector[57] backbone was PCR amplified using the primers pHERD26TgF 5′-GGGGATCCTCTAGAGTCGAC-3′ and pHERD26TgR 5′-GGG TACCGAGCTCGAATTCTTATCAGATC-3′ with Phusion Hot Start II DNA polymerase (ThermoFisher Scientific, Massachusetts, USA). The primers used in this section are shown in Supplementary Table 3. The gene PCR products containing *lasB* were then cloned into the pHERD26T PCR-amplified plasmid backbone and transformed in NEB 5-alpha competent *E. coli* strain (New England Biolabs). Plasmid constructs are shown in Supplementary Table 2. *E. coli* strains carrying pHERD26T and pHERD26T containing the *lasB* gene (pHERD26T-*lasB*) were then conjugated into *P. aeruginosa lasB* transposon mutant by triparental mating, using pRK2013 as the mobilizing plasmid[58]. Tetracycline was used at a concentration of 150 μg/ml for plasmid selection while gentamicin was used at 50 μg/ml for *E. coli* counter selection. For the expression of *lasB* in *E. coli* under the arabinose promoter, cells were grown in the presence of 0.2% of glucose (as control) or 0.2% of arabinose.

**Cell line.** Chinese hamster ovary (CHO) cells, in which NanoLuc luciferase (Nluc) is placed at the PAR-2 N terminus, were used to measure PAR-2 cleavage (Supplementary Table 2)[27]. Briefly, Nluc reporter tag was cloned in frame with the human form of PAR-2 complementary DNA and its stop codon was mutated to insert eYFP tag (Nluc-hPAR2-eYFP). A stable CHO cell line was developed using positive selection to plasmocin antibiotic. The cell line was routinely grown in Dulbecco's minimal essential medium supplemented with 1 mM sodium pyruvate, 10% fetal bovine serum, and 2.5 μg/ml plasmocin (Invitrogen/ThermoFisher, Carlsbad, CA) on Nunclon Delta Surface Cell Culture Flasks (Sigma-Aldrich, St. Louis, MO) at 37 °C in a 5% $CO_2$ in a humidified incubator.

**Mouse colonizations and bacterial supplementations.** For monocolonizations, germ-free mice received one oral gavage with *P. aeruginosa* PA14 WT or an isogenic variant lacking elastase (*lasB* mutant;$10^8$ cfu/mouse). Clean SPF mice (altered Schaedler flora-colonized)[59] co-colonized with *P. aeruginosa* received one oral gavage with *P. aeruginosa* PA14 WT, the *lasB* mutant, or *P. aeruginosa* X46.1 ($10^{10}$ cfu/mouse). For SPF experiments, protease-resistant PAR-2 mutant and C57BL/6 mice were supplemented by oral gavage with *P. aeruginosa* PA14 WT 3 times a week for 2 weeks ($10^{10}$ cfu/mouse). The presence of *Pseudomonas* in monocolonized or supplemented mice was monitored by plating feces and small intestine luminal contents on gluten-containing agar media[32] and McConkey agar media incubated for 48 h. For colonization with human microbiota, germ-free mice received one oral gavage with aspirates from the small intestine of control or CeD donors at a final concentration of $10^4$ cfu/mouse. A total of 9 aspirates with bacterial counts of at least $10^4$ cfu/ml from 5 controls and 4 CeD donors were used for mouse colonization. Bacterial load in the small intestinal aspirates was estimated by culturing on Brain Hearth Infusion agar media and McConkey agar media. Each aspirate was sufficient to colonize 2–3 mice.

**Gluten treatment.** We used a previously validated conventional sensitization protocol where mice are gavaged with a combination of cholera toxin (CT; 25 μg) and pepsin/trypsin (PT)-digested gliadin (PT-gliadin; 500 μg) weekly, over a 3-week period. Following PT-gliadin sensitization, mice were challenged by oral gavage with 2 mg of sterile gliadin (Sigma-Aldrich) dissolved in acetic acid three times a week for 2 weeks (gluten treatment). Non-sensitized control mice receiving CT and saline during the sensitization phase and acetic acid during the challenge phase were used as sham controls[24].

**PAR-2 antagonist treatment.** To study the role of PAR-2 signaling in vivo, mice monocolonized with *P. aeruginosa* PA14 WT were intraperitoneally injected every 3 days with GB83 PAR-2 antagonist (AxonMedChem, The Netherlands) dissolved in dimethyl sulfoxide (DMSO) at a final concentration 5 mg/kg for 2 weeks. Control mice received DMSO alone.

**Evaluation of enteropathy.** At the time of killing, sections of proximal small intestine were collected in 10% formalin and paraffin embedded. Using rabbit anti-human primary antibody to CD3 (1:2000; Dako; GA50361–2), CD3+ IELs were quantified in villus tips of formalin-fixed sections. Slides were examined at ×20 magnification using light microscopy in a blinded fashion. The number of CD3+ IELs per 20 enterocytes was counted from 5 randomly chosen villus tips and expressed as IELs/100 enterocytes[60]. Paraffin-embedded sections were stained with hematoxylin and eosin for histological evaluation of tissue morphology under light microscopy (Olympus, ON, Canada). Using the Image-Pro 6.3 software (Mediacybernetics, MD, USA), enteropathy was quantified in a blinded fashion by measuring 20 villus-to-crypt ratios.

**Anti-gliadin antibodies.** Antigliadin antibodies in small intestinal washes were measured by enzyme-linked immunosorbent assay as previously described[61], with some modifications. The 96-well Maxisorp round-bottom polystyrene plates (Nunc, Roskilde, Denmark) were coated with 50 μl/well of a 0.01 mg/ml solution of the gliadin extract in 0.1 M carbonate buffer (pH 9.6) or were left uncoated to serve as control wells. Wells were blocked by incubation with 1% bovine serum albumin. Serum samples were diluted at 1:100, whereas intestinal wash samples were diluted at 1:10. The samples were added at 50 μl per well in duplicates and incubated for 1 h. Each plate contained a positive control sample. After washing the wells, they were incubated with a 1:2000 dilution of either horseradish peroxidase-conjugated anti-mouse IgG (GE Healthcare, Piscataway, NJ; NA931) or IgA (Abcam, Cambridge, MA; ab97235) secondary antibodies. The antibodies used in this section are shown in Supplementary Table 4. The plates were washed, and 50 μl of developing solution was added to each well. Absorbance was measured at 450 nm after 20 min. Absorbance values were corrected for nonspecific binding by subtraction of the mean absorbance of the associated bovine serum albumin–coated control wells. The corrected values were first normalized according to the mean value of the positive control duplicate on each plate. The mean antibody level for the *lasB* group was then set as 1.0 arbitrary units, and other results were normalized accordingly.

**Quantification of proteolytic activity.** Elastolytic, glutenasic, and mucolytic activities were measured in small intestinal contents and in bacterial culture supernatants. Glutenasic activity was also determined in human duodenal biopsies. Glutenasic and mucolytic activity were measured by bioassay (1.6% agar plates) using gluten 1% (Sigma-Aldrich) or mucin from porcine stomach type III 0.5% (Sigma-Aldrich), respectively[62]. Samples were incubated in the gluten or mucin-agar plated for 14 h. Amido black was used to stain protein in the mucin bioassay. Positive proteolytic activity was determined by the presence of a halo surrounding the inoculation site on substrate-containing media. Elastase activity was analyzed using Suc-Ala3-pNa (Sigma-Aldrich) or FITC-elastin (AnaSpec) substrates. Briefly, small intestinal washes or bacterial supernatants were incubated in 50 mM Tris-HCl buffer pH 8.2 supplemented with 1 mM $CaCl_2$, 50 mM NaCl, and Triton 0.25% at 37 °C with the different substrates. Absorbance and fluorescence were measured at various time points. Units of enzyme were determined using standard curves of elastase from porcine pancreas (Sigma-Aldrich). Tryptic activity was determined using Trypsin Activity Assay Kit from Abcam following the manufacturer's recommendations (Abcam, Cambridge, UK).

**Bioluminescent imaging.** Germ-free and clean SPF mice were gavaged with $10^{10}$ cfu *P. aeruginosa* PA14 expressing a luciferase operon (*luxCDABE*) under the control of the *lasB* promoter. Two weeks after colonization, the gastrointestinal tract was removed and imaged ex vivo in a Spectrum in vivo Imaging System (IVIS, Perkin Elmer).

**16S-fluorescence in situ hybridization.** At the time of killing, sections of proximal small intestine were collected in Carnoy's fixative and paraffin embedded. The commercial 16S rRNA gene probe EUB338 (5′-GCTGCCTCCCGTAGGAGT-3′, Integrated DNA Technologies) was 5′ end-labeled with fluorochrome Cy3 and then incubated (5 ng/μl) overnight with tissues (Supplementary Table 3). The hybridization buffer contained 0.9 M NaCl, 20 mM Tris-HCl, pH 7.3, and 0.01% sodium dodecyl sulfate. Commercial NON338 (IDT) was used as negative control.

DAPI (4′,6′-diamidino-2-phenylindole) was used to stain DNA in the tissue. Image-Pro 6.3 software was used to score bacterial encroachment in the epithelium by counting bacterial signal in contact with 100 enterocytes.

**PAR-2 cleavage and expression**. Bacterial proteases were incubated with CHO cells in which NanoLuc luciferase (Nluc) is placed at the PAR-2 N terminus. Then, $1.5 \times 10^4$ cells were plated into each well of a 96-well black plate for 24 h. The cell monolayers were then washed three times with Hank's balanced salt solution (HBSS, pH 7.4; Thermofisher), and 100 ml of OptiMeM (Gibco) was added to each well and cultured for approximately 16 h. Bacteria were incubated for 16 h in OptiMEM (Gibco) at 37 °C, after which culture medium (20 μl) was added to the Nluc-expressing cell monolayers (final volume 120 μl) and incubated for 15 min at 37 °C. Supernatant was collected and centrifuged for 5 min at 12,000 rpm. Then, 80 μl of supernatant was transferred to a 96-multiwell white plate followed by the addition of 80 μl of 50-fold diluted luciferase assay substrate solution (Promega Corporation, Madison, WI). Then, the released luciferase activity was measured using Varioskan (ThermoFisher) in duplicate. Data were normalized for each strain to the luminescence observed in the WT strains.

PAR-2 expression was quantified in the small intestine of mice using the rabbit polyclonal antibody raised against amino acids 230–238 of PAR-2 (H-99) (Santa Cruz Biotechnology, Texas, USA; sc-57797). Sections were incubated with H-99 antibody diluted 1:500 overnight. A fluorophore-conjugated secondary rabbit antibody (Alexa Fluor 594 goat anti-rabbit IgG, Life Technologies) was applied (1:2000) for 1 h (Supplementary Table 4). PAR-2 expression was quantified using ImageJ and expressed as percent of positive fluorescence/total area on the epithelial surface of the villi.

**Isolation and analysis of IELs**. Small intestines were removed from mice and IELs isolated, as previously described[28]. Briefly, small intestines from mice were flushed to remove intestinal contents, Peyer's patches and mesentery were removed, intestines opened longitudinally, and cut into 3–5 mm pieces. Intestinal pieces were incubated in HBSS containing dithiothreitol for 15 min followed by 5–6 incubations in EDTA/HEPES/HBSS in a 37 °C shaker. After each 15 min of incubation, intestines were vigorously vortexed and the IELs were collected by passing the supernatants through a 40 μm cell strainer. IELs were enriched on a Percoll gradient and resuspended in RNA later for RNA extraction or in fluorescence-activated cell sorting buffer for cell staining.

Single-cell suspensions of IELs were stained with fluorochrome-labeled cell-surface antibodies to CD45 (CD45-BV421 (30-F11) BioLegend-103133), CD3 (CD3ε-PE-dazzle 594 (145-2C11) BioLegend-100348), and CD103 (CD103-PE (3E7) BioLegend-121405) diluted 1:100. The antibodies used in this section are shown in Supplementary Table 4. Dead cells were excluded using a fixable viability dye (eBioscience). Total cell counts were determined using CountBright Absolute Cell Counting Beads (ThermoFisher). Stained cells were acquired using the LSR II (BD Biosciences) and analyzed with FlowJo software (TreeStar, Ashland, OR).

**RNA expression**. For RNA extraction from whole tissue, a Tissue-Tearor Homogenizer (Biospec) was used. RNA was prepared using the RNeasy Mini Kit (Qiagen). NanoString nCounter gene expression (Mouse Inflammation Panel, 254 genes) was run according to the manufacturer's instructions (NanoString Technologies Inc.). The results obtained were analyzed with nSolver 2.5 (NanoString Technologies). Ratios built from the data were uploaded into IPA software (Qiagen) for further analysis. The network score is based on the hypergeometric distribution and is calculated with the right-tailed Fisher's exact test.

**Microbiota sequencing**. Small intestinal contents were collected and flash frozen on dry ice. DNA was extracted from samples and amplified for the hypervariable 16S rRNA gene v3 region for sequencing on the Illumina MiSeq platform (Illumina, San Diego, CA). Briefly, 341F (5′-CCTACGGGAGGCAGCAG-3′) and 518R (5′-GTATTACCGCGGCTGCTGG-3′) 16S rRNA primers were modified for adaptation to the Illumina (San Diego, CA) platform and included the addition of 6 bp unique barcodes to the reverse primer, allowing for multiplex amplification. The amount of primer used was decreased to 5 pmol each, a *Taq* polymerase (Life Technologies, Carlsbad, CA) was used for amplification, and the cycling times were changed to 30 s for each step. Products were then sequenced using the Illumina MiSeq platform. The procedure followed main guidelines to facilitate recognition of signals that represent contamination including the use of proper negative controls[63]. Processing and analysis of the data was performed using an in-house pipeline[64]. Sequences were trimmed using Cutadapt software (version 1.2.1) and aligned through the PANDAseq software (version 2.8)[65]. Operational taxonomic units (OTUs) clustered using AbundantOTU[66] were assigned taxonomy according to the Greengenes 2011 reference database[67]. OTUs observed only once across the data (i.e., singletons) were removed. A total of 2,633,880 reads were obtained with an average of 75,240.58 per sample and ranged 1219 to 155,294 per sample. Alpha-diversity was measured using Observed species, Chao1, Shannon, and Simpson index. Beta-diversity was calculated using Bray–Curtis dissimilarity and the distance matrix was used to calculate principal coordinates and non-metric multidimensional scaling (NMSD). Differences between whole bacterial communities were tested by permutational multivariate analysis of variance (PERMANOVA) using Qiime. Differences in taxon composition were

evaluated with Kruskal–Wallis test for multiple samples or Mann–Whitney test for two independent samples.

**Statistical analysis**. All the variables were analyzed with SPSS, version 18.0 (SPSS Inc., Chicago, IL). Categorical variables are expressed as numbers and percentages, and quantitative variables as means ± standard error of the mean (s.e.m.) or medians with interquartile range for parametric or nonparametric data, respectively. Normal distribution was determined by D'Agustino–Pearson omnibus normality test, Shapiro–Wilk test and Kolmogorov–Smirnov test with Dallal-Wilkinson-Lillie correction. Data are depicted as dot plots, with each dot representing an individual human or mouse, or biological replicate. The one-way analysis of variance (ANOVA) test was used to evaluate differences between more than two groups with a parametric distribution and Tukey's correction was applied. Student's *t*-test (two-tailed) was performed to evaluate the differences between two independent groups or paired samples as appropriate. Data with nonparametric distribution were evaluated with Kruskal–Wallis test for more than two groups, and Mann–Whitney test for two independent groups. Student's *t*-test and multiple *t*-tests were used by nSolver 2.5 for statistical analysis of gene expression. Spearman's rank correlation coefficient was used to measure nonparametric rank correlation. A *P* value < 0.05 was selected to reject the null hypothesis by two-tailed tests. Information regarding specific *P* values, value of *n*, and how data are presented can be found in figure legends.

**Reporting summary**. Further information on experimental design is available in the Nature Research Reporting Summary linked to this article.

## Data availability
The data presented in the manuscript are found in the NCBI database, under the Umbrella BioProject: PRJNA51889. This project contains the Nanostring Raw data deposited in Gene Expression Omnibus database GSE125983 and the sequencing data deposited in Sequence Read Archive (SRA, SRP136344).

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

## Acknowledgements

The authors thank McMaster's AGU staff for their support with germ-free mouse breeding and gnotobiotic experiments. The authors also thank the CeD volunteers who participated in this study. The work was funded by a CIHR grant MOP#142773 and a Boris Family award to E.F.V, a European Research Council (ERC- 310973 PIPE) to N.V., and a European Research executive Agency (IOF-627487-EPIMACase) to C.D. A.C. was granted a Farncombe Fellowship Award and Campbell Research Award. M.M. was supported by a CCFA Research Fellowship Award (ID: 480735). W.R. was supported by a consortium grant of the Boehringer Ingelheim Foundation.

## Author contributions

A.C. and E.F.V. conceptualized the study. A.C., J.L.M. and H.J.G. performed most of the experiments and analyzed the data. C.D., C.R. and N.V. performed in vitro experiments regarding PAR-2 receptor signaling. W.R. provided SPF protease-resistant PAR-2 mice. C.D., N.V. and W.R. give expertise in proteolytic signaling. S.P.B., M.C. and M.G.S. provided *Pseudomonas aeruginosa* strains and mutants, and valuable expertise on total intestinal microbiota experiments and analysis. X.B.Y. and A.A. analyzed anti-gliadin antibodies. B.K.C. participated in bioluminescent imaging. P.B. recruited patients and provided duodenal biopsies. C.M.S. processed duodenal biopsies. J.A.M. sent NOD-DQ8 mice. M.M. and B.J. analyzed expression of specific innate cytokines. J.C. provided expertise and bacterial strains involved on gluten metabolism. F.C. provided expertise in host-bacteria immune interactions. A.C., H.J.G. and E.F.V. wrote the manuscript. All of the authors discussed the results and assisted in the preparation of the manuscript.

## Additional information

**Competing interests:** The authors declare no competing interests.

