## [Peer Review File · Nature Communications]

Reviewers' comments:

Reviewer #1 (Remarks to the Author):

The study entitled "duodenal bacterial proteolytic activity determines sensitivity to dietary antigen through PAR-2 receptor" proposes that specific microbes in the duodenum of patients with celiac disease (CeD) produce proteases that activated protease-activated receptor 2 on the duodenal epithelium leading to an immune response that is characteristic of this disease. The authors first identify elevated levels of protease activity in the duodenal samples from CeD patients and then use a reductionist approach with a model microorganism (*Pseudomonas aeruginosa*) to determine the mechanism by which a microbial protease can illicit an CeD-like immune response. This is a novel concept for CeD pathogenesis and the authors provide solid experiments to support their conclusions. There are some issues that the authors need to address before this study is ready for publication.

Although the experimental approach the authors take is strong there are some additional tests they could do to bolster their conclusions.

- The authors could complement the mutant *P. aeruginosa* in some of their experiments? Reintroducing the *lasB* gene into the mutant bacterium and restoring the phenotype would show specificity for this gene.
- The authors use elastase activity as a proxy for proteolytic activity. Tryptic activity could also be determined in these samples. By ignoring tryptic activity the authors are missing out on a large group of proteases.
- Did the authors consider colonizing GF mice with the duodenal microbiotas from human donors?
- The authors could use Western blots to support cleavage of PAR 2.
- PAR2 agonists/antagonists could be used to support the in vitro findings.
- Can LasB be purified and used in the in vitro experiments?

Other comments.

- Line 40, the authors state that they are giving an example for the dysfunction they mention in their previous statement, however they provide a synopsis of what they believe is happening in CeD. If there is an example, please provide it. If not then this is not an example.
- Line 100, is it really proteolytic activity if you are just measuring elastase activity? I think it would be prudent to state 'elastase activity' and mention that this is a component of total proteolytic activity.
- Line 115, at this point I think it would be suitable to define what the healthy groups was. For example, individuals screened for possible GERD may not be completely considered healthy, but are definitely non-CeD.
- Line 123 glutenase and elastase activity seem to be interchangeable. It would be better to stick to one definition. Additionally, I believe the "data not shown" should be shown, even if it is in a supplementary form.
- Line 137 define D2 glutenase activity.
- Line 200, add "in the duodenum" to be more specific.
- Line 211, I think Supp Fig 3C belongs in the main article.
- Line 233, usually intestinal microbiotas are complex, so stating that mice that harbor a complex microbiota doesn't fit. "harbor a more diverse/complex intestinal microbiota" would be more accurate.
- Line 425, please state where (cell type) the expression is different.
- Line 301, "a" is missing after "As".
- Discussion, although the authors provide convincing evidence that PAR2 is involved in this process. They need to discuss the relevance of PARs 1, 3, & 4.
- For low biomass samples (such as intestinal biopsies) there is the fear that the sequence data by be from contamination. Although, I don't believe this has happened in this study it is an important issue to keep in mind. I would encourage to read de Goffau et al.'s paper in Nature microbiology (2018) and discuss how they avoided these pitfalls.

- Figure 1 e, although the Lactobacillus and Clostridium correlations are statistically significant, the correlations seems to be driven by outlying groups.

Reviewer #2 (Remarks to the Author):

The manuscript by Alberto Caminero and colleagues addresses an important aspect of the pathogenesis of the major gluten-related disorder, celiac disease. They specifically have looked at the effects of an opportunistic pathogen, *Pseudomonas aeruginosa* and correlated duodenal cleavage of gluten proteins with presence of this bacteria. They have, in their study, combined human data (looking at biopsies), in vitro data and mouse models. The studies are carefully executed by the group, which is world-wide leading in this field. The present report also rests on a previous report on related subject, published by the group in *Gastroenterology* (Caminero et al 2016).

They first looked at biopsy materials of untreated CD patients compared to healthy controls. Great gluten-degrading capacity was found. It would be re-assuring to see a statement that none of these individuals were taking proton-pump inhibitors - I could not see those data. It would further be interesting to see comments on the persistence of the deranged microbiota in their patients, is this a consequence of the malabsorption?

They thereafter changed focus to a model system with *P. aeruginosa* producing LasB elastase and found that the mutant lacked proteolytic activity. They then went on and found that this strain can induce a pro-inflammatory phenotype.

A surprising finding of theirs was that the cleavage of PAR-2 by LasB producing bacteria was responsible for the pro-inflammatory increase, without any requirement of gluten. They finally sought to combine their data in a (imperfect) mouse model for celiac disease where HLA-DQ8 mice show villous blunting after immunization with gluten, a process which is dependent on intestinal microbiota. Here, they conclude that LasB from *P. aeruginosa* enhances the gluten-driven immunopathology through gluten independent mechanisms.

I think the experiments are very well performed and the data are clearly presented. The discussion runs smoothly. The significance for celiac disease as it is seen in humans can, of course, be debated. But the present report is a very valuable addition to our understanding.

Reviewer #3 (Remarks to the Author):

The manuscript by Caminero and colleagues describe proteases expressed by *Pseudomonas aeruginosa* that cleave PAR2 to impact host immune responses, which are explored in the context of food sensitivities triggered by protein antigens (namely gluten). I feel that there is a high level of novelty in the findings to justify publication of the research presented in this manuscript.

This research presents a comprehensive study that includes duodenal biopsies from adult celiac patient cohorts to demonstrate increased glutenase activity and microbiota changes compared to healthy cohorts. Abundance of gluten-degrading *Pseudomonas aeruginosa* correlated with increased glutenase activity. Previous studies have found conflicting roles of *Pseudomonas aeruginosa*-related elastase-dependent activity via PAR2 however here Caminero and colleagues use protease-resistant PAR2 mouse models to highlight changes intraepithelial lymphocyte (IEL) counts (but not glutenase/elastase activity) in response to LasB-producing *P. aeruginosa*, thus

demonstrating the role of PAR2 activity. The approaches that are used in the paper are well described and would enable any researcher to follow should they wish to repeat the experiments if necessary.

Introduction:

Balanced introduction covering GI proteases and their roles, receptors, responses, microbe-dietary interactions, metabolic regulation of dietary gluten by pathogens and microbial proteases.

Minor point:

Discussion content.

There is conflicting literature as to the role of *Pseudomonas aeruginosa*-derived elastase in relation to PAR2 activity. I think this needs to be at least mentioned in the discussion section (potentially somewhere amidst the discussion content in line 305 onwards)

Pseudomonas aeruginosa elastase disables Proteinase-Activated Receptor 2 - (Dulon, J Respir Cell Mol Biol Vol 32. pp 411-419, 2005). <https://www.ncbi.nlm.nih.gov/pubmed/15705968>

And

<https://onlinelibrary.wiley.com/doi/full/10.1111/j.1462-5822.2008.01142.x>

I would encourage the authors to incorporate balance into the discussion by including reference in some way to these previous studies in light of the results they present.

My view is that the findings of this manuscript will lead to further investigations into this topic and will influence the field considerably in the years to come. The role of PAR2 that has been described will certainly be of high interest to those who seek to develop drugs for this target. I would recommend that the paper is published subject to the minor additions suggested in the discussion content.

Responses to Reviewers' comments:

We thank the reviewers for their insightful comments and suggestions, which we believe have significantly increased the value of this manuscript.

Reviewer #1 (Remarks to the Author):

The study entitled “duodenal bacterial proteolytic activity determines sensitivity to dietary antigen through PAR-2 receptor” proposes that specific microbes in the duodenum of patients with celiac disease (CeD) produce proteases that activated protease-activated receptor 2 on the duodenal epithelium leading to an immune *response* that is characteristic of this disease. The authors first identify elevated levels of protease activity in the duodenal samples from CeD patients and then use a reductionist approach with a model microorganism (*Pseudomonas aeruginosa*) to determine the mechanism by which a microbial protease can illicit an CeD-like immune response. This is a novel concept for CeD pathogenesis and the authors provide solid experiments to support their conclusions. There are some issues that the authors need to address before this study is ready for publication.

Although the experimental approach the authors take is strong there are some additional tests they could do to bolster their conclusions.

1- The authors could complement the mutant *P. aeruginosa* in some of their experiments? Reintroducing the *lasB* gene into the mutant bacterium and restoring the phenotype would show specificity for this gene.

Response: We thank the referee for this comment. We agree that reintroducing *lasB* gene into the *P. aeruginosa lasB* mutant bacterium is a solid way to show specificity for this gene. We have therefore, following this suggestion, performed a basic complementation of the *lasB* mutant on a plasmid. The *lasB* gene sequence was introduced into the pHERD26T plasmid either under the endogenous *P. aeruginosa* promoter (starting 300 bp upstream of start site) or under the plasmid's arabinose promoter using Gibson cloning (New England Biolabs). Plasmids of interest were introduced into *P. aeruginosa* by conjugation using *E. coli* containing the pRK2013 plasmid. Consistent with our previously presented findings, degradation of gluten by *P. aeruginosa* is abolished by deletion of *lasB*. Conjugation of *P. aeruginosa lasB* mutant with *lasB*-expressing plasmid under the endogenous *P. aeruginosa* promoter results in gluten degradation that is absent from *P. aeruginosa lasB* mutant conjugated with the empty plasmid. In addition, transformation of *E. coli* with plasmid coding for *lasB* expression under the arabinose-inducible promoter results in gluten degradation in

the presence of the inducer arabinose, but not in the presence of glucose. These results confirm *lasB* specificity against gluten proteins, and are included in Supplementary Figure 2. Additionally, we tested all of these mutants for their ability to degrade the external domain of PAR-2 *in vitro*. Complementation of the *P. aeruginosa lasB* transposon mutant with the *lasB*-expressing plasmid restored the capacity of the strain to degrade the external domain of PAR-2. In addition, transformation of *E. coli* strain (that not have capacity to release PAR-2 N-terminus peptide) with the plasmid coding for *lasB* resulted in high level PAR-2 receptor cleavage under the arabinose inducible promoter and much lower degradation when repressed by the addition of glucose (Supplementary Figure 4a). These results demonstrate the capacity of LasB to cleave the external domain of PAR-2 and are now included in the manuscript.

2- The authors use elastase activity as a proxy for proteolytic activity. Tryptic activity could also be determined in these samples. By ignoring tryptic activity the authors are missing out on a large group of proteases.

Response: We apologize for not being clear in our first submission. We are detecting an increase in proteolytic activity *against gluten proteins* (thus, this is termed “glutenasic” activity) in duodenal samples of patients with celiac disease vs non-celiac individuals. As the referee suggests, these functional activities could have different origin such as elastase- or trypsin-like. For instance, certain bacteria such as *Rothia* or *Bacillus* are able to degrade gluten proteins through subtilisin (Wei J. Am J phys 2016). Unfortunately, duodenal biopsy does not contain enough material to measure different proteolytic activities. For that reason, we targeted overall activity against gluten proteins in human duodenal biopsies. We have reviewed the paper for accuracy in the terminology used. In contrast, in animal experiments using *P. aeruginosa* WT producing LasB we are able to specifically measure elastase activity. The reason we target this activity is that we have previously shown that *P. aeruginosa* WT primarily degrades gluten proteins through LasB (an elastase-like protease). The referee’s point is well taken, and we have therefore determined tryptic activity in duodenal samples of clean SPF C57BL/6 and GF C57BL/6 and NOD/ DQ8 mice supplemented or mono-colonized with *Pseudomonas aeruginosa* PA14 or LasB KO mutant. No differences were reported in tryptic activity between mice colonized with PA14 and LasB in any mouse strain. Results are shown as supplementary data (Supplementary Figure 2, 3 and 5).

3- Did the authors consider colonizing GF mice with the duodenal microbiotas from human donors?

Response: We would like to thank the referee for this important comment. These experiments were in progress when the paper was submitted. We have colonized germ-free mice with duodenal aspirates from celiac patients and individuals undergoing endoscopy for exploratory reasons but in whom organic disease (including celiac disease and IBD) was ruled out (controls). We believe this is one of the few attempts to colonize germ-free mice using low yield sample such as small intestinal contents, as opposed to colon/fecal contents. However, this was key for our experiments given the localized lesion of celiac disease in the proximal small intestine. We found that aspirates with bacterial counts of at least 10^5 cfu/ml colonized germ-free mice. One control with very low counts did not establish a stable colonization in mice, and these experiments were not included. We thus selected a total of 9 aspirates (6 from controls and 4 from CeD patients) to colonize germ-free C57BL/6 mice. Each aspirate allowed for colonization of 2-3 mice after adjustment of colony forming units. After 3 weeks, mice were sacrificed and small intestinal content was collected to determine proteolytic activities (against gluten substrates), intraepithelial lymphocytes and intestinal microbiota composition. Small intestinal microbiota in recipient mice was profiled and 16s V3 sequencing revealed the presence of human bacteria clustered significantly by donor. No differences were reported in small intestinal microbiota composition (Supplementary Figure 6B) and diversity (Figure 6B) based on CeD diagnosis. Although individual donor-related differences were observed, pooled group data revealed that mice colonized with microbiota from CeD had higher small intestinal glutenase activity (Fig. 6C) and IEL counts (Fig. 6D) than mice colonized with control microbiota. Moreover, there was a significant correlation between glutenase activity and IELs counts (Fig. 6E). As in human duodenal biopsies, glutenase activity inversely correlated with Firmicutes and directly correlated with Proteobacteria relative abundance (Fig. 6F). Taken together, these results indicate that the small intestinal microbiota of CeD patients has an increased glutenase activity that associates with an innate immune response relevant in CeD. These results are shown now in the revised manuscript.

4- The authors could use Western blots to support cleavage of PAR 2.

Response: We considered initially this possibility but, Western-blot is not recommended to demonstrate PAR-2 activation. None of the described, and available antibodies, to detect PAR-2 work well in Western-blot (strong non-specific binding) assays. The only currently accepted way to suggest activation of the receptor using antibodies is to see its internalization (Rolland-Fourcade et al. Gut 2017), but this is not a direct demonstration of activation. We therefore, to address this point, performed immunostaining of small intestinal

tissues of NOD/DQ8 mice colonized with *Pseudomonas* PA14 and LasB for PAR-2 receptor using H-99 antibody (Santa Cruz Biotechnology: sc-5595). We found an increase of PAR-2 in NOD/DQ8 samples colonized with *Pseudomonas* PA14 WT when we compared with NOD/DQ8 mice colonized with *lasB* (Supplementary Figure 5). The data has been included in the manuscript. In agreement with that, the use of the novel PAR-2 deficient mouse directly shows that PAR2 cleavage is needed for the innate activation mediated by LasB, which has been properly expanded in the discussion section. It has previously been shown that LasB can induce PAR-2 depending signaling in a very elegant paper (Dulon et al. Am. J. Respir Cell Mol Biol 2005). However, we agree that we are not able to confirm how this activation is performed *in vivo* (for instance canonical activation vs alternative) and we acknowledge this issue as a limitation of our work. This is an interesting question that would be followed up in future experiments in collaboration with Wolfram Ruf lab.

5- PAR2 agonists/antagonists could be used to support the *in vitro* findings.

Response: This is a relevant comment based on the ability of PAR-2 antagonists to revert the functional activity of LasB inducing an innate immune activation. Thus, monocolonized mice with *P. aeruginosa* PA14 WT were intraperitoneally injected every three days with GB83 PAR-2 antagonist or DMSO at a final concentration 5 mg/kg for 2 weeks (Supplementary Figure 4A). At sacrifice, both groups showed similar amounts of bacteria (10^3 - 10^4 cfu/g of small-intestinal content) and glutenase activity in the small intestine (Supplementary Figure 4B and Figure 4C). However, *P. aeruginosa* PA14 WT-monocolonized mice treated with the GB83 PAR-2 antagonist showed a reduced amount of IEL when compared with mice treated with saline (Supplementary Figure 4D), suggesting a role for PAR-2 in the innate activation by LasB. Results are shown in the revised manuscript.

6- Can LasB be purified and used in the *in vitro* experiments?

Response: There is a commercially available LasB (Calbiochem) that we have used in some *in vitro* experiments. Commercial elastase is able to degrade elastin-FITC (data not shown) and gluten substrate (shown now in Supplementary Figure 2B). Unfortunately, the protease is not stable enough for its use in animal models and in some *in vitro* experiments. The protease is affected by freeze cycles and room temperature.

7- Other comments:

7.1- Line 40, the authors state that they are giving an example for the dysfunction they mention in their previous statement, however they provide a synopsis of what they believe is happening in CeD. If there is an example, please provide it. If not then this is not an example.

R: We apologize for the incorrect wording in Line 40 corresponding to the abstract of the manuscript. The manuscript has been now revised for accuracy.

7.2- Line 100, is it really proteolytic activity if you are just measuring elastase activity? I think it would be prudent to state 'elastase activity' and mention that this is a component of total proteolytic activity.

R: Sorry for the misinterpretation, which we have also addressed above in point 2. We are detecting increased proteolytic activity against gluten proteins ("glutenasic" activity) in duodenal samples of celiac patients' vs healthy volunteers. Many different types of proteases and peptidases can degrade gluten substrates including elastases. In our *in vivo* experiments, we used *P. aeruginosa* producing LasB. We have previously shown that *P. aeruginosa* PA14 is able to degrade gluten through LasB (elastase-like) (Caminero et al. Gastroenterology 2016). For that reason, we measured elastase-like activity and glutenasic activity in mice colonized with *P. aeruginosa* PA14 and *P. aeruginosa lasB* mutant. We have reviewed the manuscript for accuracy.

7.3- Line 115, at this point I think it would be suitable to define what the healthy groups was. For example, individuals screened for possible GERD may not be completely considered healthy, but are definitely non-CeD.

R: We are sorry for the incorrect definition of our control group. We have now defined it as non-celiac individuals undergoing endoscopy in whom organic disease was ruled out. For graphical reasons and simplicity, we use the term "control" in figures, but the term is well defined in methods and figure legends.

7.4- Line 123 glutenasic and elastase activity seem to be interchangeable. It would be better to stick to one definition. Additionally, I believe the “data not shown” should be shown, even if it is in a supplementary form.

R: Please see comments in point 2 and 7.2. We define glutenasic activity as the overall proteolytic activity against gluten substrates measured by bioassay (described in methods). Elastase activity is determined in experiments using *P. aeruginosa* as LasB is a well-defined elastase that degrade gluten substrates. Data corresponding to alpha diversity (observed species, Shannon, Simpson index) and beta diversity (Bray-Curtis, Weighted and Unweighted Unifrac) in duodenal biopsies is now displayed in Supplementary Figure 1.

7.5- Line 137 define D2 glutenasic activity.

R: D2 was changed to glutenasic activity in duodenal biopsy.

7.6- Line 200, add “in the duodenum” to be more specific.

R: Added in the manuscript.

7.7- Line 211, I think Supp Fig 3C belongs in the main article.

R: Previous supplementary Figure 3c (now Supplementary Figure 4g) belongs to a subset of experiments including Supplementary Figure 3a and 3b (now Supplementary Figure 4e-f). Figure 4 presents the same message as Supplementary Figure 4 being performed in more controlled conditions. Supplementary Figure 4 shows how supplementation of *P. aeruginosa* PA14 WT can induce an increase of IEL counts in SPF mice C57BL/6 but not in SPF PAR-2 deficient mice. However, differences in microbiota composition between SPF C57BL/6 and SPF PAR-2 deficient mice contribute to innate immune responses. To avoid secondary contributions of a possible different microbiota between types of mice, we used ex-germ free C57BL/6 and PAR-2 deficient mice monocolonized mice with *P. aeruginosa* PA14 WT. Thus, Figure 4 shows the same info using mono-colonized C57BL/6 and PAR-2 deficient mice where the phenotype is stronger and cleaner. We have explained this section better in the manuscript. We show only Figure 4 to avoid excessive panels and information for the reader in the main manuscript.

7.8- Line 233, usually intestinal microbiotas are complex, so stating that mice that harbor a complex microbiota doesn't fit. "harbor a more diverse/complex intestinal microbiota" would be more accurate.

R: We have modified the sentence for accuracy.

7.9- Line 425, please state where (cell type) the expression is different.

R: This has been corrected.

7.10- Line 301, "a" is missing after "As".

R: "A" has been added to the sentence.

7.11- Discussion, although the authors provide convincing evidence that PAR2 is involved in this process. They need to discuss the relevance of PARs 1, 3, & 4.

R: This is a good point raised by the referee with a lot of interest and controversy on the field. It is possible that other PARs such as PAR-1, -2, -3 and -4 could participate in the response. There are several examples for biologically relevant PAR1-PAR2 cross-activation (e.g. Mol Cancer Res. 2004 Jul;2(7):395-402; Arterioscler Thromb Vasc Biol. 2011 Dec;31(12):e100-6) and PAR3-PAR2 cross-activation (Blood. 2012 Jan 19;119(3):874-83.). It is important to mention that the cleavage-resistant PAR2 mutant mice still responds to cross-activation by other PARs. The use of the novel PAR2 mutant mouse directly shows that PAR-2 cleavage and no other receptor crosstalk mediates the observed effects. Discussion has been expanded to address this point.

7.12- For low biomass samples (such as intestinal biopsies) there is the fear that the sequence data by be from contamination. Although, I don't believe this has happened in this study it is an important issue to keep in mind. I would encourage to read de Goffau et al.'s paper in Nature microbiology (2018) and discuss how they avoided these pitfalls.

R: Thank you for bringing this paper to our attention. Our sequence analysis followed the main guidelines to facilitate recognition of signals that represent contamination. The methodology section has been extended and the manuscript quoted.

7.13- Figure 1 e, although the *Lactobacillus* and *Clostridium* correlations are statistically significant, the correlations seem to be driven by outlying groups.

R: We agree with the reviewer that *Pseudomonas* and *Janthinobacterium* members showed a stronger correlation with glutenase activity than Firmicutes groups. Displayed p values from *Pseudomonas* and *Janthinobacterium* survived 10% False Discovery Rate (FDR) correction ($q= 0.0005$ and 0.0008 respectively) as it is stated in Figure 1 legend. *Lactobacillus* and *Clostridium* (mainly the last group) correlations seem to be driving mainly by outliers. The small intestinal microbiota is very different between individuals making difficult to do correlations in studies with a small population. It is also important to mention that recruitment of volunteers for duodenal endoscopy is not easy. Although *Lactobacillus* and *Clostridium* are classically naturally serpin producers, we did not focus on them due to the lower correlation. In this case, the main manuscript is built in the correlation of glutenase activity with *Pseudomonas* activity, a bacterium previously described to degrade gluten efficiently through LasB.

Reviewer #2 (Remarks to the Author):

The manuscript by Alberto Caminero and colleagues addresses an important aspect of the pathogenesis of the major gluten related disorder, celiac disease. They specifically have looked at the effects of an opportunistic pathogen, *Pseudomonas aeruginosa* and correlated duodenal cleavage of gluten proteins with presence of this bacteria. They have, in their study, combined human data (looking at biopsies), in vitro data and mouse models. The studies are carefully executed by the group, which is world-wide leading in this field. The present report also rests on a previous report on related subject, published by the group in Gastroenterology (Caminero et al 2016).

They first looked at biopsy materials of untreated CD patients compared to healthy controls. Greater gluten-degrading capacity was found. It would be re-assuring to see a statement that none of these individuals were taking proton-pump inhibitors - I could not see those data. It would further be interesting to see comments on the persistence of the deranged microbiota in their patients, is this a consequence of the malabsorption?

Response: We thank the reviewer for pointing at these important statements. Consumption of proton-pump inhibitors was a factor considered in the recruitment of celiac disease patients and controls. It is well described in the literature that proton-pump inhibitors could

modify intestinal microbiota (Gut 2016; 65:740–748) and we were aware of that in the recruitment process. The information has now been added to the main manuscript. In addition, the explanation for an altered microbiota in celiac patients could depend of multiple factors including a causative role of the microbiota in the disease or a consequence of intestinal damage/inflammation. As the referee claims, malabsorption is an important factor to consider together with dietary habits. It is not easy to answer these questions in our study because a follow-up study will be needed. We are working in the lab now to elucidate the multiple factors explaining an altered microbiota in CeD patients. Here we show the role of bacterial proteases in inducing an innate immune activation relevant in CeD. However, at this point, we cannot display in the paper all the main forces driving the changes but the issue has been properly discussed in the manuscript.

They thereafter changed focus to a model system with *P. aeruginosa* producing LasB elastase and found that the mutant lacked proteolytic activity. They then went on and found that this strain can induce a pro-inflammatory phenotype.

A surprising finding of theirs was that the cleavage of PAR-2 by LasB producing bacteria was responsible for the pro-inflammatory increase, without any requirement of gluten. They finally sought to combine their data in a (imperfect) mouse model for celiac disease where HLA-DQ8 mice show villous blunting after immunization with gluten, a process which is dependent on intestinal microbiota. Here, they conclude that LasB from *P. aeruginosa* enhances the gluten-driven immunopathology through gluten independent mechanisms.

Response: We agree with the referee that animal models do not always display the same events as in humans. Our model has its limitations, as any other animal model that attempts to mimic complex human disease. The strength of the model we use is the ability to recognize gluten based on HLA transgenic expression, and the use of bacteria derived and isolated from the human intestine (Am J Pathol. 2015 Nov;185(11):2969-82). We thus believe we employ the right mouse model for the specific answer we want to address. We aimed at studying pathways that can be common or important in human disease, and we clarify and acknowledge this in the discussion.

I think the experiments are very well performed and the data are clearly presented. The discussion runs smoothly. The significance for celiac disease as it is seen in humans can, of course, be debated. But the present report is a very valuable addition to our understanding.

Response: We thank the referee for this insight. We concur fully. We have revised the claims for significance as hypotheses involving the unknown environmental triggers of celiac disease.

Reviewer #3 (Remarks to the Author):

The manuscript by Caminero and colleagues describe proteases expressed by *Pseudomonas aeruginosa* that cleave PAR2 to impact host immune responses, which are explored in the context of food sensitivities triggered by protein antigens (namely gluten). I feel that there is a high level of novelty in the findings to justify publication of the research presented in this manuscript.

This research presents a comprehensive study that includes duodenal biopsies from adult celiac patient cohorts to demonstrate increased glutenase activity and microbiota changes compared to healthy cohorts. Abundance of gluten-degrading *Pseudomonas aeruginosa* correlated with increased glutenase activity. Previous studies have found conflicting roles of *Pseudomonas aeruginosa*-related elastase-dependent activity via PAR2 however here Caminero and colleagues use protease-resistant PAR2 mouse models to highlight changes intraepithelial lymphocyte (IEL) counts (but not glutenase/elastase activity) in response to LasB-producing *P. aeruginosa*, thus demonstrating the role of PAR2 activity. The approaches that are used in the paper are well described and would enable any researcher to follow should they wish to repeat the experiments if necessary. Balanced introduction covering GI proteases and their roles, receptors, responses, microbe-dietary interactions, metabolic regulation of dietary gluten by pathogens and microbial proteases. My view is that the findings of this manuscript will lead to further investigations into this topic and will influence the field considerably in the years to come. The role of PAR2 that has been described will certainly be of high interest to those who seek to develop drugs for this target. I would recommend that the paper is published subject to the minor additions suggested in the discussion content.

Minor point:

Discussion content.

There is conflicting literature as to the role of *Pseudomonas aeruginosa*-derived elastase in relation to PAR2 activity. I think this needs to be at least mentioned in the discussion section (potentially somewhere amidst the discussion content in line 305 onwards)

Pseudomonas aeruginosa elastase disables Proteinase-Activated Receptor 2 - (Dulon, J Respir Cell Mol Biol Vol 32. pp 411–419, 2005).

<https://www.ncbi.nlm.nih.gov/pubmed/15705968> And

<https://onlinelibrary.wiley.com/doi/full/10.1111/j.1462-5822.2008.01142.x>

I would encourage the authors to incorporate balance into the discussion by including reference in some way to these previous studies in light of the results they present.

Response: We thank the referee for the enthusiasm regarding our manuscript. We agree with the referee about the conflicting literature regarding activation of PAR-2 by *P. aeruginosa*. It is possible that LasB is not the only protease in *P. aeruginosa* interacting with PAR-2 as it is mentioned in Kida et al (Cell Microbiol. 2008). However, our *in vitro* experiments (Figure 4A) suggested the specific capacity of LasB to cleave the external domain of PAR-2. We have performed additional experiments to explore the specificity of LasB against gluten peptides and PAR-2 as Referee 1 required (please see comment 1 above). Briefly, the *lasB* gene sequence was introduced into a commercial plasmid either under the endogenous *P. aeruginosa* promoter or under the plasmid's arabinose promoter and transformed into *E. coli*. Plasmids of interest were introduced into *P. aeruginosa LasB* mutant by conjugation. Genetic complementation of *P. aeruginosa lasB* mutant with *lasB*-expressing plasmid under the endogenous *P. aeruginosa* promoter resulted in degradation of gluten and the external domain of PAR-2 that was absent from *P. aeruginosa lasB* mutant conjugated with the empty plasmid. In addition, transformation of *E. coli* strain (that not have capacity to degrade gluten or PAR-2) with the plasmid coding for *lasB* resulted in degradation of gluten and PAR-2 receptor under the arabinose inducible promoter, but not in the presence of glucose (Supplementary Figure 2 and 4). These results confirm *lasB* specificity against gluten and the external domain of PAR-2 and are now included in the revised manuscript.

As mentioned by the reviewer, Dulon et al. (J Respir Cell Mol Biol 2005 Vol 32. pp 411–419) have shown that LasB disables the function of PAR2 in the respiratory tract, thereby altering the host innate defense mechanisms. In our study, we did not study how the PAR2-LasB interaction occurs and we acknowledge this as future potential work. However, Dulon et al. (J Respir Cell Mol Biol 2005 Vol 32. pp 411–419) have shown in a very elegant manner that

PAR2-LasB interaction can lead to an alternative activation of the receptor. The use of a novel PAR-2 deficient mouse in our study directly shows that PAR-2 cleavage mediates the innate activation by LasB. It is important to mention that PAR-2 resistant mutant mice still respond to cross-activation by other PARs. Thus, we conclude the role of PAR-2 cleavage in our response, independently of *how* it is activated, is mediated by a canonical or an alternative cleavage. We have included this issue in the discussion. We believe detailed description of the mechanism is beyond the scope of this paper and could be addressed in the future.

REVIEWERS' COMMENTS:

Reviewer #1 (Remarks to the Author):

The authors have gone beyond the expectations to improve their study. I believe the manuscript in its current state is acceptable for publication.